# CoLA: Exploiting Compositional Structure for Automatic and Efficient Numerical Linear Algebra

**Andres Potapczynski**[*1]  **Marc Finzi**[*2]  **Geoff Pleiss**[3,4]  **Andrew Gordon Wilson**[1]

[1]New York University, [2]Carnegie Mellon University, [3]University of British Columbia,

[4]Vector Institute

## Abstract

Many areas of machine learning and science involve large linear algebra problems, such as eigendecompositions, solving linear systems, computing matrix exponentials, and trace estimation. The matrices involved often have Kronecker, convolutional, block diagonal, sum, or product structure. In this paper, we propose a simple but general framework for large-scale linear algebra problems in machine learning, named *CoLA* (Compositional Linear Algebra). By combining a linear operator abstraction with compositional dispatch rules, CoLA automatically constructs memory and runtime efficient numerical algorithms. Moreover, CoLA provides memory efficient automatic differentiation, low precision computation, and GPU acceleration in both JAX and PyTorch, while also accommodating new objects, operations, and rules in downstream packages via multiple dispatch. CoLA can accelerate many algebraic operations, while making it easy to prototype matrix structures and algorithms, providing an appealing drop-in tool for virtually any computational effort that requires linear algebra. We showcase its efficacy across a broad range of applications, including partial differential equations, Gaussian processes, equivariant model construction, and unsupervised learning.

## 1  Introduction

The framework of automatic differentiation has revolutionized machine learning. Although the rules that govern derivatives have long been known, automatically computing derivatives was a nontrivial process that required (1) efficient implementations of base-case primitive derivatives, (2) software abstractions (autograd and computation graphs) to compose these primitives into complex computations, and (3) a mechanism for users to modify or extend compositional rules to new functions. Once libraries such as PyTorch, Chainer, Tensorflow, JAX, and others [1, 8, 30, 31, 38, 47] figured out the correct abstractions, the impact was enormous. Efforts that previously went into deriving and implementing gradients could be repurposed into developing new models.

In this paper, we automate another notorious bottleneck for ML methods: performing large-scale linear algebra (e.g. matrix solves, eigenvalue problems, nullspace computations). These ubiquitous operations are at the heart of principal component analysis, Gaussian processes, normalizing flows, equivariant neural networks, and many other applications [2, 12, 13, 17, 18, 27, 28, 34, 37, 39]. Modeling assumptions frequently manifest themselves as algebraic structure—such as diagonal dominance, sparsity, or a low-rank factorization. Given a structure (e.g., the sum of low-rank plus diagonal matrices) and a linear algebraic operation (e.g., linear solves), there is often a computational routine (e.g. the linear-time Woodbury inversion formula) with lower computational complexity than a general-purpose routine (e.g., the cubic-time Cholesky decomposition). However, exploiting

---

*Equal contribution.

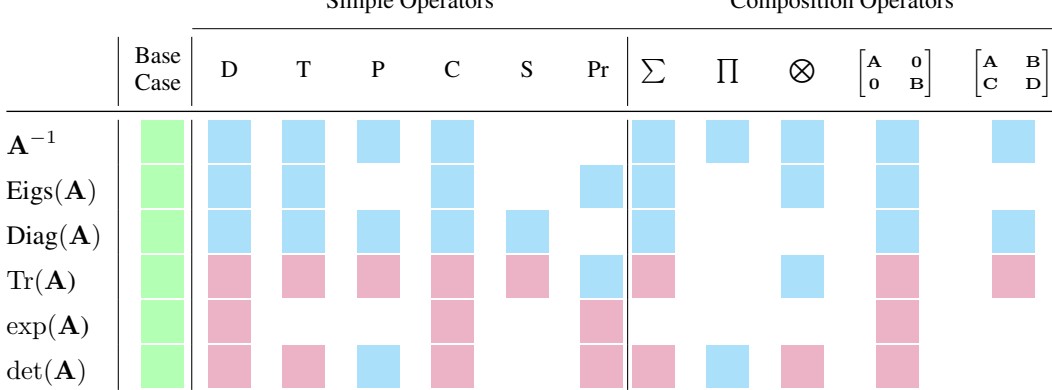

Table 1: **Many structures have explicit composition rules to exploit.** Here we show the existence of a dispatch rule ( ) that can be used to accelerate a linear algebraic operation for some matrix structure over what is possible with the dense and iterative base cases. Many combinations (shown with  ) are automatically accelerated as a consequence of other rules, since for example `Eigs` and `Diag` are used in other routines. In absence of a rule, the operation will fall back to the iterative and dense base case for each operation (shown in  ). Columns are basic linear operator types such as D: Diagonal, T: Triangular, P: Permutation, C: Convolution, S:Sparse, Pr: Projection and composition operators such as sum, product, Kronecker product, block diagonal and concatenation. All compositional rules can be mixed and matched and are implemented through multiple dispatch.

structure for faster computation is often an intensive implementation process. Rather than having an object $\mathbf{A}$ in code that represents a low-rank-plus-diagonal matrix and simply calling $\mathrm{solve}(\mathbf{A}, \mathbf{b})$, a practitioner must instead store the low-rank factor $\mathbf{F}$ as a matrix, the diagonal $\mathbf{d}$ as a vector, and implement the Woodbury formula from scratch. Implementing structure-aware routines in machine learning models is often seen as a major research undertaking. For example, a nontrivial portion of the Gaussian process literature is devoted to deriving specialty inference algorithms for structured kernel matrices [e.g. 7, 11, 19, 25, 29, 46, 52, 53, 24].

As with automatic differentiation, structure-aware linear algebra is ripe for automation. We introduce a general numerical framework that dramatically simplifies implementations efforts while achieving a high degree of computational efficiency. In code, we represent structure matrices as `LinearOperator` objects which adhere to the same API as standard dense matrices. For example, a user can call $\mathbf{A}^{-1}\mathbf{b}$ or $\mathrm{eig}(\mathbf{A})$ on any `LinearOperator` $\mathbf{A}$, and under-the-hood our framework derives a computationally efficient algorithm built from our set of compositional *dispatch rules* (see Table 1). If little is known about $\mathbf{A}$, the derived algorithm reverts to a general-purpose base case (e.g. Gaussian elimination or GMRES for linear solves). Conversely, if $\mathbf{A}$ is known to be the Kronecker product of a lower triangular matrix and a positive definite Toeplitz matrix, for example, the derived algorithm uses specialty algorithms for Kronecker, triangular, and positive definite matrices. Through this compositional pattern matching, our framework can match or outperform special-purpose implementations across numerous applications despite relying on only a small number of base `LinearOperator` types.

Furthermore, our framework offers additional novel functionality that is necessary for ML applications (see Table 2). In particular, we automatically compute gradients, diagonals, transposes and adjoints of linear operators, and we modify classic iterative algorithms to ensure numerical stability in low precision. We also support specialty algorithms, such as SVRG [23] and a novel variation of Hutchinson's diagonal estimator [22], which exploit *implicit structure* common to matrices in machine learning applications (namely, the ability to express matrices as large-scale sums amenable to stochastic approximations). Moreover, our framework is easily extensible in *both directions*: a user can implement a new linear operator (i.e. one column in Table 1), or a new linear algebraic operation (i.e. one row in Table 1). Finally, our routines benefit from GPU and TPU acceleration and apply to symmetric and non-symmetric operators for both real and complex numbers.

We term our framework *CoLA* (**Co**mpositional **L**inear **A**lgebra), which we package in a library that supports both PyTorch and JAX. We showcase the extraordinary versatility of CoLA with

a broad range of applications in Section 3.2 and Section 4, including: PCA, spectral clustering, multi-task Gaussian processes, equivariant models, neural PDEs, random Fourier features, and PDEs like minimal surface or the Schrödinger equation. Not only does CoLA provide competitive performance to specialized packages but it provides significant speedups especially in applications with compositional structure (Kronecker, block diagonal, product, etc). Our package is available at https://github.com/wilson-labs/cola.

## 2    Background and Related Work

**Structured matrices**    Structure appears throughout machine learning applications, either occurring naturally through properties of the data, or artificially as a constraint to simplify complexity. A nonexhausitve list of examples includes: (1) low-rank matrices, which admit efficient solves and determinants [54]; (2) sparse matrices, which admit fast methods for linear solves and eigenvalue problems [14, 44]; (3) Kronecker-factorizable matrices, which admit efficient spectral decompositions; (4) Toeplitz or circulant matrices, which admit fast matrix-vector products. See Section 3 and Section 4 for applications that use these structures. Beyond these explicit types, we also consider *implicit structures*, such as matrices with clustered eigenvalues or matrices with simple unbiased estimates. Though these implicit structures do not always fall into straightforward categorizations, it is possible to design algorithms that exploit their inherent properties (see Section 3.3).

**Iterative matrix-free algorithms**    Unlike direct methods, which typically require dense instantiations of matrices, matrix-free algorithms only access matrices through routines that perform matrix-vector multiples (MVMs) [e.g. 44]. The most common matrix-free algorithms—such as conjugate gradients, GMRES, Lanczos and Arnoldi iteration—fall under the category of Krylov subspace methods, which iteratively apply MVMs to refine a solution until a desired error tolerance is achieved. Though the rate of convergence depends on the conditioning or spectrum of the matrix, the number of iterations required is often much less than the size of the matrix. These algorithms often provide significant computational speedups for structured matrices that admit sub-quadratic MVMs (e.g. sparse, circulant, Toeplitz, etc.) or when using accelerated hardware (GPUs or TPUs) designed for efficient parallel MVMs [e.g. 10, 20, 51].

**Multiple dispatch**    Popularized by `Julia` [6], multiple dispatch is a functional programming paradigm for defining type-specific behaviors. Under this paradigm, a given function (e.g. `solve`) can have multiple definitions, each of which are specific to a particular set of input types. A base-case definition `solve[LinearOperator]` would use a generic matrix-vector solve algorithm (e.g. Gaussian elimination or GMRES), while a type-specific definition (e.g. `solve[Sum]`, for sums of matrices) would use a special purpose algorithm that makes use of the subclass' structure (e.g. SVRG, see Section 3.3). When a user calls $\mathrm{solve}(\mathbf{A}, \mathbf{b})$ at runtime, the *dispatcher* determines which definition of `solve` to use based on the types of $\mathbf{A}$ and $\mathbf{b}$. Crucially, dispatch rules can be written for compositional patterns of types. For example, a `solve[Sum[LowRank, Diagonal]]` function will apply the Woodbury formula to a `Sum` operator that composes `LowRank` and `Diagonal` matrices. (In contrast, under an inheritance paradigm, one would need to define a specific `SumOfLowRankAndDiagonal` sub-class that uses the Woodbury formula, rather than relying on the composition of general purpose types.)

**Existing frameworks for exploiting structure**    Achieving fast computations with structured matrices is often a manual effort. Consider for example the problems of second order/natural gradient optimization, which require matrix solves with (potentially large) Hessian/Fisher matrices. Researchers have proposed tackling these solves with matrix-free methods [33], diagonal approximations [e.g. 4], low-rank approximations [e.g. 42], or Kronecker-factorizable approximations [34]. Despite their commonality—relying on structure for fast solves—all methods currently require different implementations, reducing interoperability and adding overhead to experimenting with new structured approximations. As an alternative, there are existing libraries like SciPy Sparse [50], Spot [49], PyLops [41], or GPyTorch [20], which offer a unified interface for using matrix-free algorithms with any type of structured matrices. A user provides an efficient MVM function for a given matrix and then chooses the appropriate iterative method (e.g. conjugate gradients or GMRES) to perform the desired operation (e.g. linear solve). With these libraries, a user can adapt to different structures simply by changing the MVM routine. However, this increased interoperability comes at the cost of efficiency, as the iterative routines are not optimal for every type of structure. (For example, Kronecker products admit efficient inverses that are asymptotically faster than conjugate

gradients; see Figure 1.) Moreover, these libraries often lack modern features (e.g. GPU acceleration or automatic differentiation) or are specific to certain types of matrices (see Table 2).

| Package | GPU Support | Autograd | Non-symmetric Matrices | Complex Numbers | Randomized Algorithms | Composition Rules |
|---|---|---|---|---|---|---|
| `Scipy Sparse` | ✗ | ✗ | ✓ | ✓ | ✗ | ✗ |
| `PyLops` | ✳ | ✳ | ✓ | ✓ | ✗ | ✗ |
| `GPyTorch` | ✓ | ✓ | ✗ | ✗ | ✗ | ✗ |
| `CoLA` | ✓ | ✓ | ✓ | ✓ | ✓ | ✓ |

Table 2: Comparison of scalable linear algebra libraries. `PyLops` only supports propagating gradients through vectors but not through the linear operator's parameters. Moreover, `PyLops` has limited GPU support through CUPY, but lacks support for PyTorch, JAX or TensorFlow which are necessary for modern machine learning applications.

## 3 CoLA: Compositional Linear Algebra

We now discuss all the components that make CoLA. In Section 3.1 we first describe the core MVM based `LinearOperator` abstraction, and in Section 3.2 we discuss our core compositional framework for identifying and automatically exploiting structure for fast computations. In Section 3.3, we highlight how CoLA exploits structure frequently encountered in ML applications beyond well-known analytic formulae (e.g. the Woodbury identity). Finally, in Section 3.4 we present CoLA's machine learning-specific features, like automatic differentiation, support for low-precision, and hardware acceleration.

### 3.1 Deriving Linear Algebraic Operations Through Fast MVMs

Borrowing from existing frameworks like `Scipy Sparse`, the central object of our framework is the `LinearOperator`: a linear function on a finite dimensional vector space, defined by how it acts on vectors via a matrix-vector multiply $\text{MVM}_A : \mathbf{v} \mapsto \mathbf{A}\mathbf{v}$. While this function has a matrix representation for a given basis, we do not need to store or compute this matrix to perform a MVM. Avoiding the dense representation of the operator saves memory and often compute.

Some basic examples of `LinearOperators` are: unstructured `Dense` matrices, which are represented by a 2-dimensional array and use the standard MVM routine $[\mathbf{A}\mathbf{v}]_i = \sum_{j=1} A_{ij} v_j$; `Sparse` matrices, which can be represented by key/value arrays of the nonzero entries with the standard CSR-sparse MVM routine; `Diagonal` matrices, which are represented by a 1-dimensional array of the diagonal entries and where the MVM is given by $[\text{Diag}(\mathbf{d})\mathbf{v}]_i = d_i v_i$; `Convolution` operators, which are represented by a convolutional filter array and where the MVM is given by $\text{Conv}(\mathbf{a})\mathbf{v} = \mathbf{a} * \mathbf{v}$ ; or JVP operators—the Jacobian represented implicitly through an autograd Jacobian Vector Product—represented by a function and an input $\mathbf{x}$ and where the MVM is given by $\text{Jacobian}(f, \mathbf{x})\mathbf{v} = \text{JVP}(f, \mathbf{x}, \mathbf{v})$. In CoLA, each of these examples are sub-classes of the `LinearOperator` superclass.

Through the `LinearOperator`'s MVM, it is possible to derive other linear algebraic operations. As a simple example, we obtain the dense representation of the `LinearOperator` by calling $\text{MVM}(\mathbf{e}_1)$, ..., $\text{MVM}(\mathbf{e}_N)$, on each unit vector $\mathbf{e}_i$. We now describe several key operations supported by our framework, some well-established, and others novel to CoLA.

**Solves, eigenvalue problems, determinants, and functions of matrices** As a base case for larger matrices, CoLA uses *Krylov subspace methods* (Section 2, Appendix C) for many matrix operations. Specifically, we use GMRES [43] for matrix solves and Arnoldi [3] for finding eigenvalues, determinants, and functions of matrices. Both of these algorithms can be applied to any non-symmetric and/or complex linear operator. When `LinearOperators` are annotated with additional structure (e.g. self-adjoint, positive semi-definite) we use more efficient Krylov algorithms like MINRES, conjugate gradients, and Lanczos (see Section 3.2). As stated in Section 2, these algorithms are

| | $\Pi_i^M \mathbf{A}_i$ | $\sum_i^M \mathbf{A}_i$ | BlockDiag$(\mathbf{A}, \mathbf{B})$ | Kron$(\mathbf{A}, \mathbf{B})$ |
|---|---|---|---|---|
| MVM $(\tau)$ | $\sum_i \tau_i$ | $\sum_i \tau_i$ | $\tau_A + \tau_B$ | $\tau_A N_B + N_A \tau_B$ |
| Solve $(s)$ | $\sum_i \kappa_i \tau_i \log \frac{M}{\epsilon}$ | $(1 + \kappa/M)\tau \log \frac{1}{\epsilon}$ | $s_A + s_B$ | $s_A N_B + N_A s_B$ |
| Eigs $(E)$ | $\tau \log \frac{M}{\epsilon} \Pi_i \kappa_i$ | $(1 + \kappa/M)\tau \log \frac{1}{\epsilon}$ | $E_A + E_B$ | $E_A + E_B$ |

Table 3: **CoLA selects the best rates for each operation or structure combination.** Asymptotic runtimes resulting from dispatch rules on compositional linear operators in our framework. Listed operations are matrix vector multiplies, linear solves, and eigendecomposition. Here $\epsilon$ denotes error tolerance. For a given operator of size $N \times N$, we denote $\tau$ as its MVM cost, $s$ its linear solve cost, $E$ its eigendecomposition cost and $\kappa$ its condition number. A lower script indicates to which matrix the operation belongs to.

matrix free (and thus memory efficient), amenable to GPU acceleration, and asymptotically faster than dense methods. See Section C.2 for a full list of Krylov methods used by CoLA.

**Transposes and complex conjugations** In alternative frameworks like `Scipy Sparse` a user must manually define a transposed MVM $\mathbf{v} \mapsto \mathbf{A}^{\mathsf{T}}\mathbf{v}$ for linear operator objects. In contrast, CoLA uses a novel autograd trick to derive the transpose from the core MVM routine. We note that $\mathbf{A}^{\mathsf{T}}\mathbf{v}$ is the vector-Jacobian product (VJP) of the vector $\mathbf{v}$ and the Jacobian $\partial \mathbf{A}\mathbf{w}/\partial \mathbf{w}$. Thus, the function `transpose(A)` returns a `LinearOperator` object that uses $\text{VJP}(\text{MVM}_{\mathbf{A}}, \mathbf{0}, \mathbf{v})$ as its MVM. We extend this idea to Hermitian conjugates, using the fact that $\mathbf{A}^{*}\mathbf{v} = (\overline{\mathbf{A}})^{\mathsf{T}}\mathbf{v} = \overline{(\mathbf{A}^{\mathsf{T}}\overline{\mathbf{v}})}$.

**Other operations** In Section 3.3 we outline how to stochastically compute diagonals and traces of operators with MVMs, and in Section 3.4 we discuss a novel approach for computing memory-efficient derivatives of iterative methods through MVMs.

**Implementation** CoLA implements all operations (`solve`, `eig`, `logdet`, `transpose`, `conjugate`, etc.) following a functional programming paradigm rather than as methods of the `LinearOperator` object. This is not a minor implementation detail: as we demonstrate in the next section, it is crucial for the efficiency and compositional power of our framework.

## 3.2 Beyond Fast MVMs: Exploiting Explicit Structure Using Composition Rules

While the GMRES algorithm can compute solves more efficiently than corresponding dense methods such as the Cholesky decomposition, especially with GPU parallelization and preconditioning, it is not the most efficient algorithm for many `LinearOperators`. For example, if $\mathbf{A} = \text{Diag}(\mathbf{a})$, then we know that $\mathbf{A}^{-1} = \text{Diag}(\mathbf{a}^{-1})$ without needing to solve a linear system. Similarly, solves with triangular matrices can be inverted efficiently through back substitution, and solves with circulant matrices can be computed efficiently in the Fourier domain $\text{Conv}(\mathbf{a}) = \mathcal{F}^{-1}\text{Diag}(\mathcal{F}\mathbf{a})\mathcal{F}$ (where $\mathcal{F}$ is the Fourier transform linear operator). We offer more examples in Table 1 (left).

As described in Section 2, we use multiple dispatch to implement these special case methods. For example, we implement the `solve[Diagonal]`, `solve[Triangular]`, and `solve[Circulant]` dispatch rules using the efficient routines described above. If a specific `LinearOperator` subclass does not have a specific `solve` dispatch rule then we default to the base-case `solve` rule using GMRES. This behaviour also applies to other operations, such as `logdet`, `eig`, `diagonal`, etc.

The dispatch framework makes it easy to implement one-off rules for the basic `LinearOperator` sub-classes described in Section 3.1. However, its true power lies in the use of compositional rules, which we describe below.

**Compositional Linear Operators** In addition to the base `LinearOperator` sub-classes (e.g. `Sparse`, `Diagonal`, `Convolution`), our framework provides mechanisms to compose multiple `LinearOperators` together. Some frequently used compositional structures are Sum ($\sum_i \mathbf{A}_i$), Product ($\Pi_i \mathbf{A}_i$), Kronecker ($\mathbf{A} \otimes \mathbf{B}$), KroneckerSum ($\mathbf{A} \oplus \mathbf{B}$), BlockDiag $[\mathbf{A}, 0; \ 0, \overline{\mathbf{B}}]$ and Concatenation $[\mathbf{A}, \mathbf{B}]$. Each of these compositional `LinearOperators` are defined by (1) the base `LinearOperator` objects to be composed, and (2) a corresponding MVM routine, which is typically written in terms of the MVMs of the composed `LinearOperators`. For example, $\text{MVM}_{\text{Sum}} = \mathbf{v} \mapsto \sum_i \text{MVM}_i(\mathbf{v})$, where $\text{MVM}_i$ are the MVM routines for the component `LinearOperators`.

Dispatch rules for compositional operators are especially powerful. For example, consider `Kronecker` products where we have the rule $(\mathbf{A} \otimes \mathbf{B})^{-1} = \mathbf{A}^{-1} \otimes \mathbf{B}^{-1}$. Though simple, this rule yields highly efficient routines for numerous structures. For example, suppose we want to solve $(\mathbf{A} \otimes \mathbf{B} \otimes \mathbf{C})\mathbf{x} = \mathbf{b}$ where $\mathbf{A}$ is dense, $\mathbf{B}$ is diagonal, and $\mathbf{C}$ is triangular. From the rules, the solve would be split over the product, using GMRES for $\mathbf{A}$, diagonal inversion for $\mathbf{B}$, and forward substitution for $\mathbf{C}$. This breakdown is much more efficient than the base case (GMRES with `MVM`$_{\texttt{Kron}}$).

When exploited to their full potential, these composition rules provide both asymptotic speedups (shown in Table 3) as well as runtime improvements on real problems across practical sizes (shown in Figure 1). Splitting up the problem with composition rules yields speedups in surprising ways even in the fully iterative case. To illustrate, consider one large CG solve with the matrix power $\mathbf{B} = \mathbf{A}^n$; in general, the runtime is upper-bounded by $O(n\tau\sqrt{\kappa^n}\log\frac{1}{\epsilon})$, where $\tau$ is the time for a MVM with $\mathbf{A}$, $\kappa$ is the condition number of $\mathbf{A}$, and $\epsilon$ is the desired error tolerance. However, splitting the product via a composition rule into a sequence of solves has a much smaller upper-bound of $O(n\tau\sqrt{\kappa}\log\frac{n}{\epsilon})$. We observe this speedup in the solving the Bi-Poisson PDE shown in Figure 1(b).

**Additional flexibly and efficiency via parametric typing**   A crucial advantage of multiple dispatch is the ability to write simple special rules for compositions of specific operators. While a general purpose `solve`[Sum] method (SVRG; see next section) yields efficiency over the GMRES base case, it is not the most efficient algorithm when the `Sum` operator is combining a `LowRank` and a `Diagonal` operator. In this case, the Woodbury formula would be far more efficient. To account for this, CoLA allows for dispatch rules on *parametric types*; that is, the user defines a `solve[Sum[LowRank, Diagonal]]` dispatch rule that is used if the `Sum` operator is specifically combining a `LowRank` and a `Diagonal` linear operator. Coding these rules without multiple dispatch would require specialty defining sub-classes like `LowRankPlusDiagonal` over the `LinearOperator` object, increasing complexity and hampering extendibility.

**Decoration/annotation operators**   Finally, we include several *decorator* types that annotate existing `LinearOperator`s with additional structure. For example, we define `SelfAdjoint` (Hermetian/symmetric), `Unitary` (orthonormal), and `PSD` (positive semi-definite) operators, each of which wraps an existing `LinearOperator` object. None of these decorators define a specialty MVM; however, these decorators can be used to define dispatch rules for increased efficiency. For example `solve[PSD]` can use conjugate gradients rather than GMRES, and `solve[PSD[Tridiagonal]]` can use the linear time tridiagonal Cholesky decomposition [see e.g., 21, Sec. 4.3.6].

**Taken together**   Our framework defines 16 base linear operators, 5 compositional linear operators, 6 decoration linear operators, and roughly 70 specialty dispatch rules for `solve`, `eig`, and other operations. (See Table 1 for a short summary and Appendix A for a complete list of rules.) We note that these numbers are relatively small compared with existing solutions yet—as we demonstrate in Section 4— these operators and dispatch rules are sufficient to match or exceed performance of specialty implementations in numerous applications. Finally, we note that CoLA is extensible by users in *both directions*. A user can write their own custom dispatch rules, either to (1) define a new `LinearOperator` and special dispatch rules for it, or (2) to define a new algebraic operation for all `LinearOperator`s, and crucially this requires no changes to the original implementation.

### 3.3   Exploiting Implicit Structure in Machine Learning Applications

So far we have discussed *explicit* matrix structures and composition rules for which there are simple analytic formulas easily found in well-known references [e.g. 21, 44, 48]. However, current large systems—especially those found in machine learning— often have *implicit structure* and special properties that yield additional efficiencies. In particular, many ML problems give rise to linear operators composed of large summations which are amenable to stochastic algorithms. Below we outline two impactful general purpose algorithms used in CoLA to exploit this implicit structure.

**Accelerating iterative algorithms on large sums with SVRG**   Stochastic gradient descent (SGD) is widely used for optimizing problems with very large or infinite sums to avoid having to traverse the full dataset per iteration. Like Monte Carlo estimation, SGD is very quick to converge to a few decimal places but very slow to converge to higher accuracies. When an exact solution is required on a problem with a finite sum, the stochastic variance reduced gradient (SVRG) algorithm [23] is much more compelling, converging on strongly convex problems (and many others) at an exponential rate, with runtime $O((1 + \kappa/M)\log\frac{1}{\epsilon})$ where $\kappa$ is the condition number and $\epsilon$ is the desired accuracy.

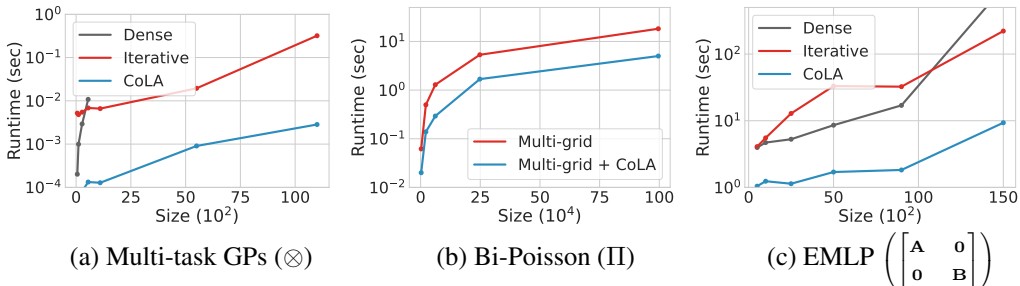

(a) Multi-task GPs ($\otimes$)  (b) Bi-Poisson ($\Pi$)  (c) EMLP $\left( \begin{bmatrix} \mathbf{A} & \mathbf{0} \\ \mathbf{0} & \mathbf{B} \end{bmatrix} \right)$

Figure 1: **Empirically, our composition rules yield the best runtimes** across applications consisting of linear operators with different structures (more application details in Section 4). We plot mean runtime (over 3 repetitions) for different methods (dense, iterative and ours (CoLA)) against the size of the linear operator. **(a)** Computing solves on a multi-task GP problem [7] for a linear operator having Kronecker structure $\mathbf{K}_T \otimes \mathbf{K}_X$, where $\mathbf{K}_T$ is a kernel matrix containing the correlation between the tasks and $\mathbf{K}_X$ is a RBF kernel on the data. For this experiment we used a synthetic Gaussian dataset which has dimension $D = 33$, $N = 1$K and we used $T = 11$ tasks. **(b)** Computing solves on the 2-dimensional Bi-Poisson PDE problem for the composition of the Laplacian operator $\Delta$ composed with itself on grid of sizes up to $N = 1000^2$. We use CG with a multi-grid $\alpha$SA preconditioner [9] to solve the linear system required in this application. **(c)** Finding the nullspace of an equivariant MLP of a linear operator having block diagonal structure. Here, NullF refers to the iterative nullspace finder algorithm detailed in [16]. We ran a 5-node symmetric operator $S(5)$ as done in [16] with MLP sizes up to 15K. See Appendix D for further details.

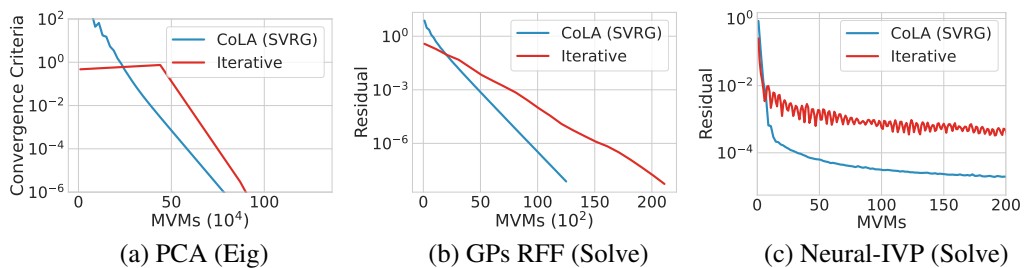

(a) PCA (Eig)  (b) GPs RFF (Solve)  (c) Neural-IVP (Solve)

Figure 2: **CoLA exploits the sum structure of linear operators through stochastic routines**. **(a)** Eigenvalue convergence criteria against number of MVMs for computing the first principal component on Buzz ($N = 430$K, $D = 77$) using VR-PCA [45]. **(b)** Solve relative residual against number of MVMs for a random Fourier features (RFFs) approximation [40] to a RBF kernel with $J = 1$K features on Elevators ($N = 12.5$K, $D = 18$). **(c)** Solve relative residual against number of MVMs when applying Neural-IVP [17] to the 2-dimensional wave equation equation as done in [17]. See Appendix D for further details.

When the condition number and the number of elements in the sum is large, SVRG becomes a desirable alternative even to classical deterministic iterative algorithms such as CG or Lanczos whose runtimes are bounded by $O(\sqrt{\kappa} \log \frac{1}{\epsilon})$. Figure 2 shows the impact of using SVRG to exploit the structure of different linear operators that are composed of large sums.

**Stochastic diagonal and trace estimation with reduced variance**    Another case where we exploit implicit structure is when estimating the trace or the diagonal of a linear operator. While collecting the diagonal for a dense matrix is a trivial task, it is a costly algorithm for an arbitrary `LinearOperator` defined only through its `MVM`—it requires computing $\text{Diag}(\mathbf{A}) = \sum_{i=1}^{N} e_i \odot \mathbf{A} e_i$ where $\odot$ is the Hadamard (elementwise) product. If we need merely an approximation or unbiased estimate of the diagonal (or the sum of the diagonal), we can instead perform stochastic diagonal estimation [22] $\overline{\text{Diag}}(\mathbf{A}) = \frac{1}{n} \sum_{j=1}^{n} z_j \odot \mathbf{A} z_j$ where the $z_j \in \mathbb{R}^N$ are any randomly sampled probe vectors with covariance $I$. We extend this randomized estimator to use randomization both in the probes, and random draws from a sum when $\mathbf{A} = \sum_{i=1}^{M} \mathbf{A}_i$:

$$\overline{\text{Diag}} \left( \sum_{i=1}^{M} \mathbf{A}_i \right) := \sum_{ij} z_{ij} \odot \mathbf{A}_i z_{ij}.$$

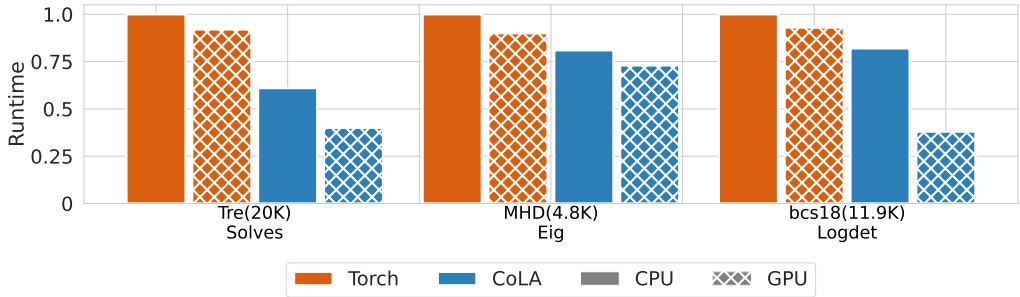

Figure 3: **For sufficiently large problems, switching from dense to iterative algorithms provides consistent runtime reductions, especially on a GPU, where matrix multiplies can be effectively parallelized.** We plot the ratio between the runtime of a linear algebra operation using CoLA or PyTorch on different hardware (CPU and GPU) divided by the runtime of using PyTorch CPU. For the linear solves, we use the matrix market sparse operator Trefethen; for the eigenvalue estimation, we use the matrix market sparse operator mhd4800b and, finally, for the log determinant computation, we use the matrix market sparse operator bcsstk18. We provide additional details in Section D.4.

In Section B.1 we derive the variance of this estimator and we show that it converges faster than the base Hutchinson estimator when applied Sum structures. We validate empirically this analysis in Figure 5.

### 3.4 Automatic Differentiation and Machine Learning Readiness

**Memory efficient auto-differentiation** In ML applications, we want to backpropagate through operations like $A^{-1}$, Eigs($A$), Tr($A$), exp($A$), $\log \det(A)$. To achieve this, in CoLA we define a novel concept of the gradient of a LinearOperator which we detail in Appendix B. For routines like GMRES, SVRG, and Arnoldi, we utilize a custom backward pass that does not require backproagating through the iterations of these algorithms. This custom backward pass results in substantial memory savings (the computation graph does not have to store the intermediate iterations of these algorithms), which we demonstrate in Appendix B (Figure 6).

**Low precision linear algebra** By default, all routines in CoLA support the standard `float32` and `float64` precisions. Moreover, many CoLA routines also support `float16` and `bfloat16` half precision using algorithmic modifications for increased stability. In particular, we use variants of the GMRES, Arnoldi, and Lanczos iterations that are less susceptible to instabilities that arise through orthogonalization [44, Ch. 6] and we use the half precision variant of conjugate gradients introduced by Maddox et al. [32]. See Appendix C for further details.

**Multi framework support and GPU/TPU acceleration** CoLA is compatible with both PyTorch and JAX. This compatibility not only makes our framework *plug-and-play* with existing implemented models, but it also adds GPU/TPU support, differentiating it from existing solutions (see Table 2). CoLA's iterative algorithms are the class of linear algebra algorithms that benefit most from hardware accelerators as the main bottleneck of these algorithms are the MVMs executed at each iteration, which can easily be parallelized on hardware such as GPUs. Figure 3 empirically shows the additional impact of hardware accelerators across different datasets and linear algebra operations.

## 4 Applications

We now apply CoLA to an extensive list of applications showing the impact, value and broad applicability of our numerical linear algebra framework, as illustrated in Figure 4. This list of applications encompasses PCA, linear regression, Gaussian processes, spectral clustering, and partial differential equations like the Schrödinger equation or minimal surface problems. In contrast to Section 3 (Figure 1 & Figure 2), the applications presented here have a basic structure (sparse, vector-product, etc) but not a compositional structure (Kronecker, product, block diagonal, etc). We choose these applications due to their popularity and heterogeneity (the linear operators have different properties: self-adjoint, positive definite, symmetric and non-symmetric), and to show that CoLA

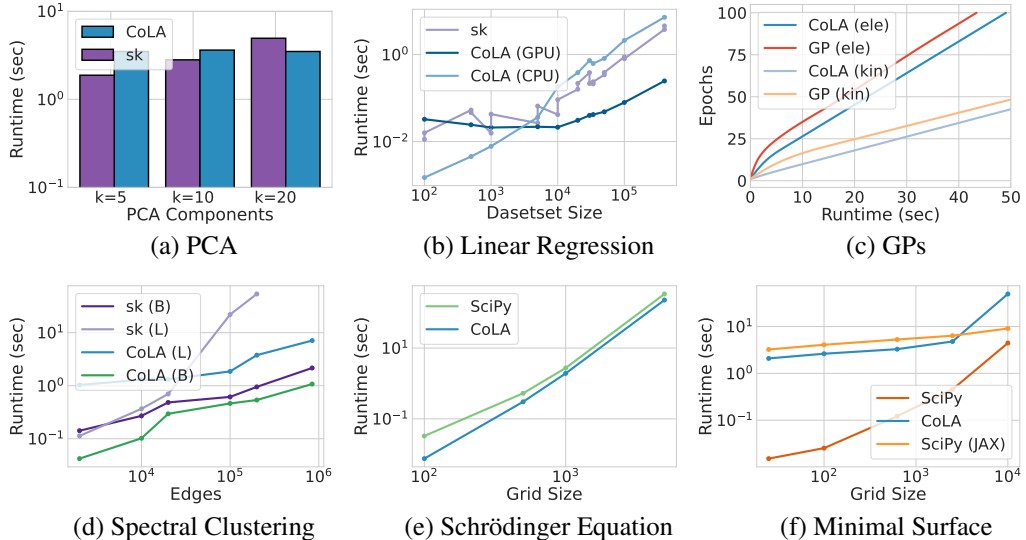

Figure 4: **CoLA is easily applied to numerous applications with competitive performance**. Here sk: sklearn, GP: GPyTorch and the tuple $(N, D)$ denotes dataset size and dimensionality. **(a)**: Runtime for PCA decomposition on Buzz (437.4K, 77). **(b)**: Linear regression runtime on Song (386.5K, 90), where we run CoLA on both GPU and CPU. **(c)**: Training efficiency (measure in epochs) on exact GP inference on Elevators (14K, 18) and Kin (20K, 8) on GPU. **(d)**: Spectral clustering runtime on a citations graph (cit-HepPh) consisting on 34.5K nodes and 842K edges. sk(L) denotes sklearn's implicitly restarted Lanczos implementation and sk(A) denotes sklearn's LOBPCG with an algebraic multi-graph preconditioner (PyAMG) [5, 26]. CoLA(L) denotes our Lanczos implementation and CoLA(B) our LOBPCG implementation. **(e)**: Runtimes for finding the smallest eigenfunctions expanding grids of a Schrödinger equation with an expanding finite difference grid. **(f)**: Runtimes for solving the minimal surface equation via root finding on expanding grids. Here SciPy utilizes the `ARPACK` package, a highly-optimized Fortran implementation of the Arnoldi iteration, while SciPy JAX (the SciPy version integrated with JAX) and CoLA utilize python Arnoldi implementations. Appendix D expands on the experimental details.

performs in any application. We compare against several well-known libraries, sometimes providing runtime improvements but other times performing equally. This is remarkable as our numerical framework does not specialize in any of those applications (like `GPyTorch`) nor does it rely on Fortran implementations of high-level algorithms (like `sklearn` or `SciPy`). Below we describe each of the applications found in Figure 4.

**Principal Component Analysis**    PCA is a classical ML technique that finds the directions in the data that capture the most variance. PCA can be performed by computing the right singular vectors of $\mathbf{X} \in \mathbb{R}^{N \times D}$. When the number of data points $N$ is very large, stochastic methods like SVRG in VR-PCA [45] can accelerate finding the eigenvectors over SVD or Lanczos, as shown in Figure 2(a).

**Spectral Clustering**    Spectral clustering [36] finds clusters of individual nodes in a graph by analyzing the graph Laplacian $\mathbf{L} = \mathbf{D} - \mathbf{W}$ where $\mathbf{D}$ denotes a diagonal matrix containing the degree of the nodes and $\mathbf{W}$ the weights on the edges between nodes. This problem requires finding the smallest $k$ eigenvectors of $\mathbf{L}$. We run this experiment on the high energy physics arXiv paper citation graph (cit-HepPh).

**Gaussian processes**    GPs are flexible nonparametric probabilistic models where inductive biases are expressed through a covariance (kernel) function. At its core, training a GP involves computing and taking gradients of the log determinant of a kernel $\log |\mathbf{K}|$ and of a quadratic term $\mathbf{y}^T \mathbf{K}^{-1} \mathbf{y}$ (where $\mathbf{y}$ is the vector of observations).

**Schrödinger Equation**    In this problem we characterize the spectrum of an atom or molecule by finding the eigenspectrum of a PDE operator in a Schrodinger equation $\mathbf{H}\psi = E\psi$. After discretizing $\psi$ to a grid, we compute the smallest eigenvalues and eigenvectors of the operator $\mathbf{H}$ which for this experiment is non-symmetric as we perform a compactfying transform.

**Minimal Surface**   Here we solve a set of nonlinear PDEs with the objective of finding the surface that locally minimizes its area under given boundary constraints. When applied to the graph of a function, the PDE can be expressed as $f(z) = (1 + z_x^2)z_{yy} - 2z_x z_y z_{xy} + (1 + z_y^2)z_{xx} = 0$ and solved by root finding on a discrete grid. Applying Newton-Raphson, we iteratively solve the non-symmetric linear system $z \leftarrow z - \mathbf{J}^{-1}f(z)$ where $\mathbf{J}$ is the Jacobian of the PDE operator.

**Bi-Poisson Equation**   The Bi-Poisson equation $\Delta^2 u = \rho$ is a linear boundary value PDE relevant in continuum mechanics, where $\Delta$ is the Laplacian. When discretized using a grid, the result is a large symmetric system to be solved. We show speedups from the product structure in Figure 1(b).

**Neural PDEs**   Neural networks show promise for solving high dimensional PDEs. One approach for initial value problems requires advancing an ODE on the neural network parameters $\theta$, where $\dot{\theta} = \mathbf{M}(\theta)^{-1}F(\theta)$ where $\mathbf{M}$ is an operator defined from Jacobian of the neural network which decomposes as the sum over data points $\mathbf{M} = \frac{1}{N}\sum_i \mathbf{M}_i$ and where $F$ is determined by the governing dynamics of the PDE [15, 17]. By leveraging the sum structure with SVRG, we provide further speedups over Finzi et al. [17] as shown in Figure 2(c).

**Equivariant Neural Network Construction**   As shown in [16], constructing the equivariant layers of a neural network for a given data type and symmetry group is equivalent to finding the nullspace of a large linear equivariance constraint $\mathbf{C}\mathbf{v} = \mathbf{0}$, where the constraint matrix $\mathbf{C}$ is highly structured, being a block diagonal matrix of concatenated Kronecker products and Kronecker sums of sparse matrices. In Figure 1(c) we show the empirical benefits of exploiting this structure.

## 5   Discussion

We have presented the CoLA framework for structure-aware linear algebraic operations in machine learning applications and beyond. Building on top of dense and iterative algorithms, we leverage explicit composition rules via multiple dispatch to achieve algorithmic speedups across a wide variety of practical applications. Algorithms like SVRG and a novel variation of Hutchinson's diagonal estimator exploit implicit structure common to large-scale machine learning problems. Finally, CoLA supports many features necessary for machine learning research and development, including memory efficient automatic differentiation, multi-framework support of both JAX and PyTorch, hardware acceleration, and lower precision.

While structure exploiting methods are used across different application domains, domain knowledge often does not cross between communities. We hope that our framework brings these disparate communities and ideas together, enabling rapid development and reducing the burden of deploying fast methods for linear algebra at scale. Much like how automatic differentiation simplified and accelerated the training of machine learning models—with custom autograd functions as the exception rather than the rule—CoLA has the potential to streamline scalable linear algebra.

## Acknowledgements

This work is supported by NSF Award 1922658, NSF CAREER IIS-2145492, BigHat Biosciences, Capital One, and an Amazon Research Award.

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

## Appendix Outline

This Appendix is organized as follows:

- In Appendix A we describe various dispatch rules including the base rules, the composition rules and rules derived from other rules.
- In Appendix B we provide an extended discussion of several noteworthy features of CoLA, such as doubly stochastic estimators and memory-efficient autograd implementation.
- In Appendix C we include pseudo-code on various of the iterative methods incorporated in CoLA and discuss modifications to improve lower precision performance.
- In Appendix D we expand on the details of the experiments in the main text.

## A   Dispatch Rules

We now present the linear algebra identities that we use to exploit structure in CoLA.

### A.1   Core Functions

#### A.1.1   Inverses

We incorporate several identities for the compositional operators: product, Kronecker product, block diagonal and sum. For product we have $(\mathbf{AB})^{-1} = (\mathbf{B}^{-1}\mathbf{A}^{-1})$ and for Kronecker product we have $(\mathbf{A} \otimes \mathbf{B})^{-1} = \mathbf{A}^{-1} \otimes \mathbf{B}^{-1}$. In terms of block compositions we have the following identities:

$$\begin{bmatrix} \mathbf{A} & \mathbf{0} \\ \mathbf{0} & \mathbf{D} \end{bmatrix}^{-1} = \begin{bmatrix} \mathbf{A}^{-1} & \mathbf{0} \\ \mathbf{0} & \mathbf{D}^{-1} \end{bmatrix} \quad \text{and} \quad \begin{bmatrix} \mathbf{A} & \mathbf{B} \\ \mathbf{0} & \mathbf{D} \end{bmatrix}^{-1} = \begin{bmatrix} \mathbf{A}^{-1} & -\mathbf{A}^{-1}\mathbf{B}\mathbf{D}^{-1} \\ \mathbf{0} & \mathbf{D}^{-1} \end{bmatrix}$$

$$\begin{bmatrix} \mathbf{A} & \mathbf{B} \\ \mathbf{C} & \mathbf{D} \end{bmatrix}^{-1} = \begin{bmatrix} \mathbf{I} & -\mathbf{A}^{-1}\mathbf{B} \\ \mathbf{0} & \mathbf{I} \end{bmatrix} \begin{bmatrix} \mathbf{A} & \mathbf{0} \\ \mathbf{0} & \mathbf{D} - \mathbf{C}\mathbf{A}^{-1}\mathbf{B} \end{bmatrix}^{-1} \begin{bmatrix} \mathbf{I} & \mathbf{0} \\ -\mathbf{C}\mathbf{A}^{-1} & \mathbf{I} \end{bmatrix}$$

Finally, for sum we have the Woodbury identity and its variants. Namely, for Woodbury we have

$$(\mathbf{A} + \mathbf{UBV})^{-1} = \mathbf{A}^{-1} - \mathbf{A}^{-1}\mathbf{U}\left(\mathbf{B}^{-1} + \mathbf{VA}^{-1}\mathbf{U}\right)^{-1}\mathbf{VA}^{-1},$$

the Kailath variant where

$$(\mathbf{A} + \mathbf{BC})^{-1} = \mathbf{A}^{-1} - \mathbf{A}^{-1}\mathbf{B}\left(\mathbf{I} + \mathbf{CA}^{-1}\mathbf{B}\right)\mathbf{CA}^{-1}$$

and the rank one update via the Sherman-Morrison formula

$$(\mathbf{A} + \mathbf{bc}^{\mathsf{T}})^{-1} = \mathbf{A}^{-1} - \frac{1}{1 + \mathbf{c}^{\mathsf{T}}\mathbf{Ab}}\mathbf{A}^{-1}\mathbf{bc}^{\mathsf{T}}\mathbf{A}^{-1}.$$

Besides the compositional operators, we have some rules for some special operators. For example, for $\mathbf{A} = \mathtt{Diag}\left(\mathbf{a}\right)$ we have $\mathbf{A}^{-1} = \mathtt{Diag}\left(\mathbf{a}^{-1}\right)$. Also, if $\mathbf{Q}$ is unitary then $\mathbf{Q}^{-1} = \mathbf{Q}^*$ or if $\mathbf{Q}$ is orthonormal then $\mathbf{Q}^{-1} = \mathbf{Q}^{\mathsf{T}}$.

#### A.1.2   Eigendecomposition

We now assume that the matrices in this section are diagonalizable. That is, $\mathtt{Eigs}\left(\mathbf{A}\right) = \mathbf{\Lambda_A}, \mathbf{V_A}$, where $\mathbf{A} = \mathbf{V_A}\mathbf{\Lambda_A}\mathbf{V_A}^{-1}$. In terms of the compositional operators, there is not a general rule for product or sum. However, for the Kronecker product we have $\mathtt{Eigs}(\mathbf{A} \otimes \mathbf{B}) = \mathbf{\Lambda_A} \otimes \mathbf{\Lambda_B}, \mathbf{V_A} \otimes \mathbf{V_B}$ and for the Kronecker sum we have $\mathtt{Eigs}(\mathbf{A} \oplus \mathbf{B}) = \mathbf{\Lambda_A} \oplus \mathbf{\Lambda_B}, \mathbf{V_A} \otimes \mathbf{V_B}$. Finally, for block diagonal we have

$$\mathtt{Eigs}\left(\begin{bmatrix} \mathbf{A} & \mathbf{0} \\ \mathbf{0} & \mathbf{D} \end{bmatrix}\right) = \begin{bmatrix} \mathbf{\Lambda_A} & \mathbf{0} \\ \mathbf{0} & \mathbf{\Lambda_D} \end{bmatrix}, \begin{bmatrix} \mathbf{V_A} & \mathbf{0} \\ \mathbf{0} & \mathbf{V_D} \end{bmatrix}.$$

### A.1.3 Diagonal

As a base case, if we need to compute $\mathrm{Diag}\,(\mathbf{A})$ for a general matrix $\mathbf{A}$ we may compute each diagonal element by $\mathbf{e}_i^\mathsf{T}\mathbf{A}\mathbf{e}_i$. Additionally, if $\mathbf{A}$ is large enough we switch to randomized estimation $\mathrm{Diag}(\mathbf{A}) \approx (\mathbf{Z} \odot \mathbf{A}\mathbf{Z})\mathbf{1}/N$ with $\mathbf{Z} \sim \mathcal{N}(0,1)^{d \times N}$ where $N$ is the number of samples used to approximate the diagonal. In terms of compositional operators, we have that for sum $\mathrm{Diag}\,(\mathbf{A} + \mathbf{B}) = \mathrm{Diag}\,(\mathbf{A}) + \mathrm{Diag}\,(\mathbf{B})$. For Kronecker product we have $\mathrm{Diag}(\mathbf{A} \otimes \mathbf{B}) = \mathrm{vec}\big(\mathrm{Diag}(\mathbf{A})\mathrm{Diag}(\mathbf{B})^\mathsf{T}\big)$ and for Kronecker sum $\mathrm{Diag}(\mathbf{A} \oplus \mathbf{B}) = \mathrm{vec}\big(\mathrm{Diag}\,(\mathbf{A})\,\mathbf{1}^\mathsf{T} + \mathbf{1}\mathrm{Diag}\,(\mathbf{B})^\mathsf{T}\big)$. Finally, for block composition we have

$$\mathrm{Diag}\left(\begin{bmatrix} \mathbf{A} & \mathbf{B} \\ \mathbf{C} & \mathbf{D} \end{bmatrix}\right) = [\mathrm{Diag}(\mathbf{A}), \mathrm{Diag}(\mathbf{D})].$$

### A.1.4 Transpose / Adjoint

As explained in Section 3.1, as a base case we have an automatic procedure to compute the transpose or adjoint of any operator $\mathbf{A}$ via autodiff. However, we also incorporate the following rules. For sum we have $(\mathbf{A} + \mathbf{B})^* = \mathbf{A}^* + \mathbf{B}^*$ and $(\mathbf{A} + \mathbf{B})^\mathsf{T} = \mathbf{A}^\mathsf{T} + \mathbf{B}^\mathsf{T}$. For product we have $(\mathbf{A}\mathbf{B})^* = \mathbf{B}^*\mathbf{A}^*$ and $(\mathbf{A}\mathbf{B})^\mathsf{T} = \mathbf{B}^\mathsf{T}\mathbf{A}^\mathsf{T}$. For Kronecker product we have $(\mathbf{A} \otimes \mathbf{B})^* = \mathbf{A}^* \otimes \mathbf{B}^*$ and $(\mathbf{A} \otimes \mathbf{B})^\mathsf{T} = \mathbf{A}^\mathsf{T} \otimes \mathbf{B}^\mathsf{T}$. For the Kronecker sum we have $(\mathbf{A} \oplus \mathbf{B})^* = \mathbf{A}^* \oplus \mathbf{B}^*$ and $(\mathbf{A} \oplus \mathbf{B})^\mathsf{T} = \mathbf{A}^\mathsf{T} \oplus \mathbf{B}^\mathsf{T}$. In terms of block composition we have

$$\left(\begin{bmatrix} \mathbf{A} & \mathbf{B} \\ \mathbf{C} & \mathbf{D} \end{bmatrix}\right)^* = \begin{bmatrix} \mathbf{A}^* & \mathbf{C}^* \\ \mathbf{B}^* & \mathbf{D}^* \end{bmatrix} \quad \text{and} \quad \left(\begin{bmatrix} \mathbf{A} & \mathbf{B} \\ \mathbf{C} & \mathbf{D} \end{bmatrix}\right)^\mathsf{T} = \begin{bmatrix} \mathbf{A}^\mathsf{T} & \mathbf{C}^\mathsf{T} \\ \mathbf{B}^\mathsf{T} & \mathbf{D}^\mathsf{T} \end{bmatrix}.$$

Finally for the annotated operators we have the following rules. $\mathbf{A}^* = \mathbf{A}$ if $\mathbf{A}$ is self-adjoint and $\mathbf{A}^\mathsf{T} = \mathbf{A}$ if $\mathbf{A}$ is symmetric.

### A.1.5 Pseudo-inverse

As a base case, if we need to compute $\mathbf{A}^+$, we may use $\mathrm{SVD}\,(\mathbf{A}) = \mathbf{U}, \mathbf{\Sigma}, \mathbf{V}$ and therefore set $\mathbf{A}^+ = \mathbf{U}\mathbf{\Sigma}^+\mathbf{V}^*$, where $\mathbf{\Sigma}^+$ inverts the nonzero diagonal scalars. If the size of $\mathbf{A}$ is too large, then we may use randomized SVD. Yet, it is uncommon to simply want $\mathbf{A}^+$, usually we want to solve a least-squares problem and therefore we can use solvers that are not as expensive to run as SVD. For the compositional operators we have the following identities. For product $(\mathbf{A}\mathbf{B})^+ = \big(\mathbf{A}^+\mathbf{A}\mathbf{B}\big)^+ \big(\mathbf{A}\mathbf{B}\mathbf{B}^+\big)^+$ and for Kronecker product we have $(\mathbf{A} \otimes \mathbf{B})^+ = \mathbf{A}^+ \otimes \mathbf{B}^+$. For block diagonal we have

$$\left(\begin{bmatrix} \mathbf{A} & \mathbf{0} \\ \mathbf{0} & \mathbf{D} \end{bmatrix}\right)^+ = \begin{bmatrix} \mathbf{A}^+ & \mathbf{0} \\ \mathbf{0} & \mathbf{D}^+ \end{bmatrix}.$$

Finally, we have some identities that are mathematically trivial but that are necessary when recursively exploiting structure as that would save computation. For example, if $\mathbf{Q}$ is unitary we know that $\mathbf{Q}^+ = \mathbf{Q}$ and similarly when $\mathbf{Q}$ is orthonormal. If $\mathbf{A}$ is self-adjoint, then $\mathbf{A}^+ = \mathbf{A}^{-1}$ and also if it is symmetric and PSD.

## A.2 Derived Functions

Interestingly, the previous core functions allow us to derive multiple rules from the previous ones. To illustrate, we have that $\mathrm{Tr}\,(\mathbf{A}) = \sum_i \mathrm{Diag}\,(\mathbf{A})_i$. Additionally, if $\mathbf{A}$ is PSD we have that $f\,(\mathbf{A}) = \mathbf{V_A} f\,(\mathbf{\Lambda_A}) \mathbf{V_A}^{-1}$ and if $\mathbf{A}$ is both symmetric and PSD then $f\,(\mathbf{A}) = \mathbf{V_A} f\,(\mathbf{\Lambda_A}) \mathbf{V_A}^\mathsf{T}$. where in both cases we used $\mathrm{Eigs}\,(\mathbf{A}) = \mathbf{\Lambda_A}, \mathbf{V_A}$. Some example functions for PSD matrices are $\mathrm{Sqrt}\,(\mathbf{A}) = \mathbf{V_A}\mathbf{\Lambda_A}^{1/2}\mathbf{V_A}^{-1}$ or $\mathrm{Log}\,(\mathbf{A}) = \mathbf{V_A} \log \mathbf{\Lambda_A}\mathbf{V_A}^{-1}$. Which also this rules allow us to define $\mathrm{LogDet}\,(\mathbf{A}) = \mathrm{Tr}\,(\mathrm{Log}\,(\mathbf{A}))$.

## A.3 Other matrix identities

We emphasize that there are a myriad more matrix identities that we do not intentionally include such as $\mathrm{Tr}(\mathbf{A} + \mathbf{B}) = \mathrm{Tr}(\mathbf{A}) + \mathrm{Tr}(\mathbf{B})$ or $\mathrm{Tr}(\mathbf{A}\mathbf{B}) = \mathrm{Tr}(\mathbf{B}\mathbf{A})$ when $\mathbf{A}$ and $\mathbf{B}$ are squared. These

additional cases are not part of our dispatch rules as either they are automatically computed from other rules (as in the first example) or they do not yield any computational savings (as in the second example).

# B Features in CoLA

## B.1 Doubly stochastic diagonal and trace estimation

**Singly Stochastic Trace Estimator** Consider the traditional stochastic trace estimator:

$$\overline{\text{Tr}}[\text{Base}](\mathbf{A}) = \tfrac{1}{n} \sum_{j=1}^{n} \mathbf{z}_j^{\mathsf{T}} \mathbf{A} \mathbf{z}_j \tag{1}$$

with each $\mathbf{z}_j \sim \mathcal{N}(\mathbf{0}, \mathbf{I}_D)$ where $\mathbf{A}$ is a $D \times D$ matrix. When $\mathbf{A}$ is itself a sum $\mathbf{A} = \tfrac{1}{m} \sum_{i=1}^{m} \mathbf{A}_i$, we can expand the trace as $\overline{\text{Tr}}[\text{Base}](\mathbf{A}) = \tfrac{1}{mn} \sum_{j=1}^{n} \sum_{i=1}^{m} \mathbf{z}_j^{\mathsf{T}} \mathbf{A}_i \mathbf{z}_j$, with probe variables shared across elements of the sum.

Consider the quadratic form $Q := \mathbf{z}^{\mathsf{T}} \mathbf{A} \mathbf{z}$, which for Gaussian random variables has a cumulant generating function of $K_Q(t) = \log \mathbb{E}[e^{tQ}] = -\tfrac{1}{2} \log \det(\mathbf{I} - 2t\mathbf{A})$. From the generating function we can derive the mean and variance of this estimator: $\mathbb{E}[Q] = K_Q'(0) = \text{Tr}(\mathbf{A})$ and $\text{Var}[Q] = K_Q''(0) = 2\text{Tr}(\mathbf{A}^2)$. Since $\overline{\text{Tr}}[\text{Base}](\mathbf{A})$ is a sum of independent random draws of $Q$, we see:

$$\mathbb{E}\big[\overline{\text{Tr}}[\text{Base}](\mathbf{A})\big] = \text{Tr}(\mathbf{A}) \quad \text{and} \quad \text{Var}\big[\overline{\text{Tr}}[\text{Base}](\mathbf{A})\big] = \frac{2}{n} \text{Tr}(\mathbf{A}^2). \tag{2}$$

**Doubly Stochastic Trace Estimator** For the doubly stochastic estimator, we choose probe variables which are sampled independently for each element of the sum:

$$\overline{\text{Tr}}[\text{Sum}](\mathbf{A}) = \tfrac{1}{nm} \sum_{j=1}^{n} \sum_{i=1}^{m} \mathbf{z}_{ij}^{\mathsf{T}} \mathbf{A}_i \mathbf{z}_{ij}. \tag{3}$$

Separating out the elements of the sum, we can write the estimator as $\overline{\text{Tr}}[\text{Sum}](\mathbf{A}) = \tfrac{1}{n} \sum_{j=1}^{n} R_j$ where $R_j$ are independent random samples of the value $R = \tfrac{1}{m} \sum_{i=1}^{m} \mathbf{z}_i^{\mathsf{T}} \mathbf{A}_i \mathbf{z}_i$. The cumulant generating function is merely $K_R(t) = \sum_{i=1}^{m} K_{Q_i}(t/m)$ where $Q_i = \mathbf{z}^{\mathsf{T}} \mathbf{A}_i \mathbf{z}$. Taking derivatives we find that,

$$\mathbb{E}[R] = K_R'(0) = \tfrac{1}{m} \sum_{i=1}^{m} \text{Tr}(\mathbf{A}_i) = \text{Tr}(\mathbf{A}), \tag{4}$$

$$\text{Var}[R] = K_R''(0) = \tfrac{1}{m^2} \sum_{i=1}^{m} 2\text{Tr}(\mathbf{A}_i^2) = \tfrac{2}{m} \text{Tr}(\tfrac{1}{m} \sum_{i=1}^{m} \mathbf{A}_i^2) \tag{5}$$

Assuming bounded moments on $\mathbf{A}_i$, then both $\mathbf{A} = \tfrac{1}{m} \sum_i \mathbf{A}_i$ and $S(\mathbf{A}) = \tfrac{1}{m} \sum_i \mathbf{A}_i^2$ will converge to fixed values as $m \to \infty$. Given that $\overline{\text{Tr}}[\text{Sum}](\mathbf{A}) = \tfrac{1}{n} \sum_{j=1}^{n} R_j$, we can now write the mean and variance of the doubly stochastic estimator:

$$\mathbb{E}\big[\overline{\text{Tr}}[\text{Sum}](\mathbf{A})\big] = \text{Tr}(\mathbf{A}) \quad \text{and} \quad \text{Var}\big[\overline{\text{Tr}}[\text{Sum}](\mathbf{A})\big] = \frac{2}{mn} \text{Tr}(S(\mathbf{A})). \tag{6}$$

As the error of the estimator can be bounded by the square root of the variance, showing that while the error for $\overline{\text{Tr}}[\text{Base}]$ is $O(1/\sqrt{n})$ (even when applied to sum structures), whereas the error for $\overline{\text{Tr}}[\text{Sum}]$ is $O(1/\sqrt{nm})$, a significant asymptotic variance reduction.

The related stochastic diagonal estimator

$$\overline{\text{Diag}}[\text{Sum}](\mathbf{A}) = \tfrac{1}{nm} \sum_{j=1}^{n} \sum_{i=1}^{m} \mathbf{z}_{ij} \odot \mathbf{A}_i \mathbf{z}_{ij}. \tag{7}$$

achieves the same $O(1/\sqrt{nm})$ convergence rate, though we omit this derivation for brevity as it is follows the same steps.

In Figure 5 we empirically how our doubly stochastic diagonal estimator outperforms the standard Hutchinson estimator.

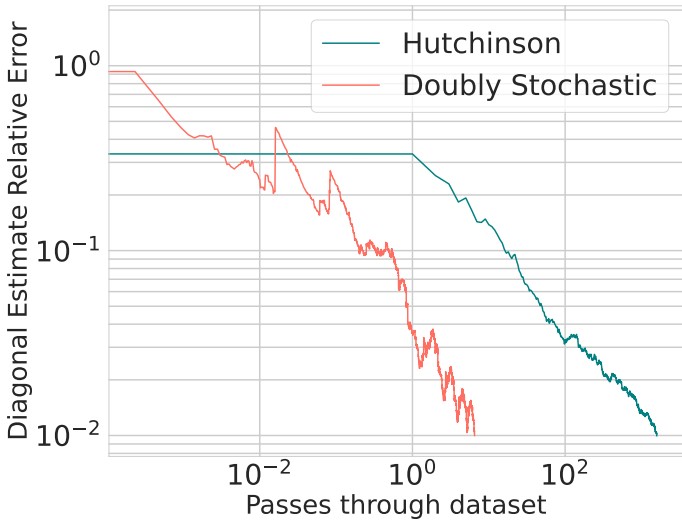

Figure 5: **Improved convergence of doubly stochastic diagonal estimator**. Convergence of our doubly stochastic diagonal estimator in evaluating the diagonal of the UCI *Buzz* empirical covariance matrix (batch size = 100). Shown is the relative error of the estimate vs the number of passes through the $n$ data points of the dataset. Our diagonal estimator has lower variance and converges faster than the standard Hutchinson estimator.

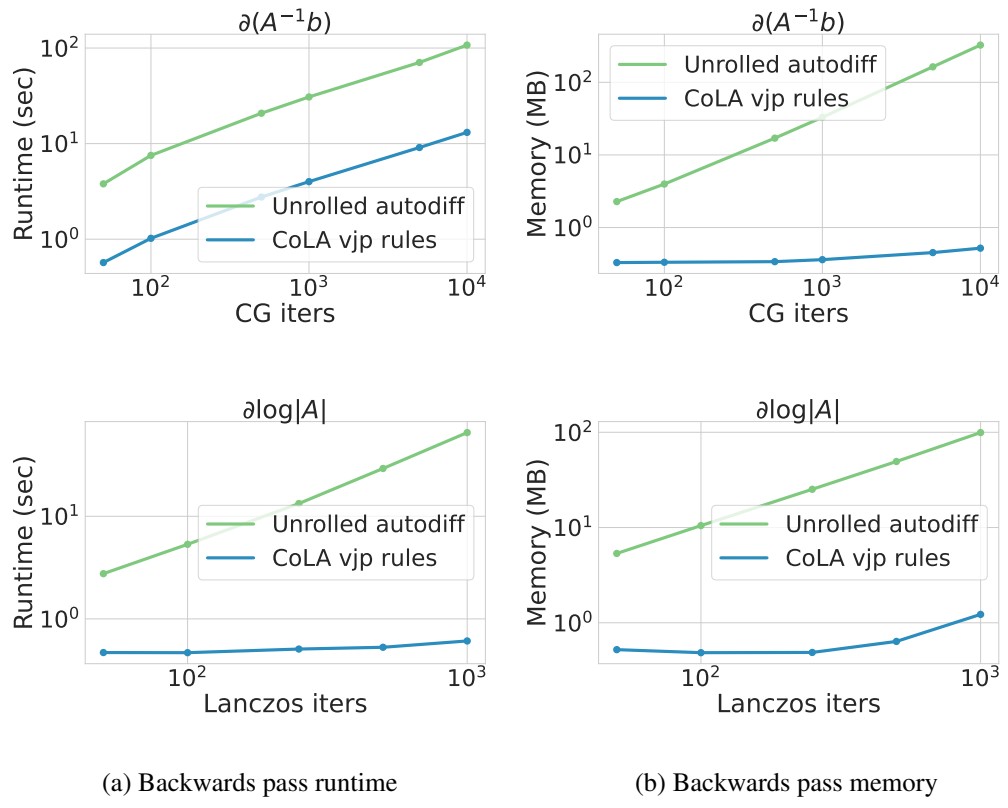

(a) Backwards pass runtime          (b) Backwards pass memory

Figure 6: **Our autograd rules allow for fast and memory efficient backpropagation**. For two different linear algebra operations $\mathbf{A}_{\boldsymbol{\theta}}^{-1}\mathbf{b}$ and $\log|\mathbf{A}_{\boldsymbol{\theta}}|$, we show the runtime and peak memory utilization required to compute the derivatives as we increase the size of the problem. In all plots, we compare CoLA's autograd rules against the autograd default of backpropagating through each iteration of the solver (unrolled autodiff). Notably, using the custom autograd rules allows us to save substantial memory and runtime when performing the backwards pass.

### B.2 Autograd rules for iterative algorithms

For machine learning applications, we want to seamlessly interweave linear algebra operations with automatic differentiation. The most basic strategy is to simply let the autograd engine trace through the operations and backpropagate accordingly. However, when using iterative methods like conjugate gradients or Lanczos, this naive approach is extremely memory inefficient and, for problems with many iterations, the cost can be prohibitive (as seen in Figure 6). However, the linear algebra operations corresponding to inverse, eigendecomposition and trace estimation have simple closed form derivatives which we can implement to avoid the prohibitive memory consumption and reduce runtime.

Simply put, for an operation like $f = \texttt{CGSolve}$, $\texttt{CGSolve}(\mathbf{A}, \mathbf{b}) = \mathbf{A}^{-1}\mathbf{b}$ we must define a Vector Jacobian Product: $\texttt{VJP}(f, (\mathbf{A}, \mathbf{b}), \mathbf{v}) = \left(\mathbf{v}^\intercal \frac{\partial f}{\partial \mathbf{A}}, \mathbf{v}^\intercal \frac{\partial f}{\partial \mathbf{b}}\right)$. However, for matrix-free linear operators, we cannot afford to store the dense matrix $\mathbf{A}$, and thus neither can we store the gradients with respect to each of its elements! Instead we must (recursively) consider how the linear operator was constructed in terms of its differentiable arguments. In other words, we must flatten the tree structure of possibly nested differentiable arguments into a vector: $\theta = \texttt{flatten}[\mathbf{A}]$. For example for $\mathbf{A} = \texttt{Kron}\big(\texttt{Diag}(\theta_1), \texttt{Conv}(\theta_2)\big)$, $\texttt{flatten}[\mathbf{A}] = [\theta_1, \theta_2]$. From this perspective, we consider $\mathbf{A}$ as a container or tree of its arguments $\theta$, and define $v^\intercal \frac{\partial f}{\partial \mathbf{A}} := \texttt{unflatten}[v^\intercal \frac{\partial f}{\partial \theta}]$ which coincides with the usual definition for dense matrices. Applying to inverses, we can now write a simple VJP:

$$\mathbf{v}^\intercal \frac{\partial f}{\partial \mathbf{A}} = \texttt{unflatten}\big[\texttt{VJP}\big(\theta \mapsto \texttt{unflatten}(\theta)\mathbf{A}^{-1}\mathbf{b}, \theta, \mathbf{A}^{-1}\mathbf{v}\big)\big] \tag{8}$$

for $\mathbf{v}^\intercal \frac{\partial f}{\partial \theta} = \mathbf{v}^\intercal (\mathbf{A}^{-1})^\intercal (\partial_\theta \mathbf{A}_\theta) \mathbf{A}^{-1}\mathbf{b}$, and we will adopt this notation below for brevity. Doing so gives a memory cost which is constant in the number of solver iterations, and proportional to the memory used in the forward pass. Below we list the autograd rules for some of the iterative routines that we implement in CoLA with their VJP definitions.

1. $\mathbf{y} = \texttt{Solve}(\mathbf{A}, \mathbf{b}) : \quad \mathbf{w}^\intercal \frac{\partial \mathbf{y}}{\partial \theta} = -(\mathbf{A}^{-1}\mathbf{w})^\intercal (\partial_\theta \mathbf{A}_\theta)(\mathbf{A}^{-1}\mathbf{b})$

2. $\boldsymbol{\lambda}, \mathbf{V} = \texttt{Eigs}(\mathbf{A}) : \quad \mathbf{w}^\intercal \frac{\partial \boldsymbol{\lambda}}{\partial \theta} = \mathbf{w}^\intercal \texttt{Diag}\big(\mathbf{V}^{-1}(\partial_\theta \mathbf{A}_\theta)\mathbf{V}\big)$

3. $\boldsymbol{\lambda}, \mathbf{V} = \texttt{Eigs}(\mathbf{A}) : \quad \mathbf{w}^\intercal \frac{\partial \mathbf{v}_i}{\partial \theta} = \mathbf{w}^\intercal (\lambda_i \mathbf{I} - \mathbf{A})^+ \partial_\theta \mathbf{A}_\theta \mathbf{v}_i$

4. $y = \log |\mathbf{A}| : \quad \frac{\partial y}{\partial \theta} = \texttt{Tr}\big(\mathbf{A}^{-1}\partial_\theta \mathbf{A}_\theta\big)$

5. $\mathbf{y} = \texttt{Diag}(\mathbf{A}) : \quad \mathbf{w}^\intercal \frac{\partial \mathbf{y}}{\partial \theta} = \mathbf{w}^\intercal \texttt{Diag}\big(\partial_\theta \mathbf{A}_\theta\big)$

In Figure 6 we show the practical benefits of our autograd rules. We take gradients of different linear solves $\mathbf{A}_\theta^{-1}\mathbf{b}$ that were derived using conjugate gradients (CG), where each solve required an increasing number of CG iterations.

## C  Algorithmic Details

In this section we expand upon three different points introduced in the main paper. For the first point we argue why SVRG leads to gradients with reduced variants. For the second points we display all the iterative methods that we use as base algorithms in CoLA. Finally, for the third point we expand upon CoLA's strategy for dealing with the different numerical precisions that we support.

### C.1  SVRG

In simplest form, SVRG [23] performs gradient descent with the varianced reduced gradient

$$\mathbf{w} \leftarrow \mathbf{w} - \eta(g_i(\mathbf{w}) - g_i(\mathbf{w}_0) + g(\mathbf{w}_0)) \tag{9}$$

where $g_i$ represents the stochastic gradient evaluated at only a single element or minibatch of the sum, and $g(\mathbf{w}_0)$ is the full batch gradient evaluated at the anchor point $\mathbf{w}_0$ which is recomputed at the end of each epoch with an updated anchor.

With different loss functions, we can use this update rule to solve symmetric or non-symmetric linear systems, to compute the top eigenvectors or even find the nullspace of a matrix. Despite the fact that the corresponding objectives are not strongly convex in the last two cases, it has been shown that

|  | Symmetric Solve $\mathbf{A}\mathbf{w} = \mathbf{b}$ | Top-$k$ Eigenvectors $\mathbf{A}\mathbf{W} = \mathbf{W}\boldsymbol{\Lambda}$ | Nullspace $\mathbf{A}\mathbf{W} = 0$ |
|---|---|---|---|
| $g_i(\mathbf{w})$ | $\mathbf{A}_i\mathbf{w} - \mathbf{b}$ | $-\mathbf{A}_i\mathbf{W} + \mathbf{W}\mathbf{W}^\intercal\mathbf{W}$ [55] | $\mathbf{A}_i\mathbf{W}$ [16] |

Table 4: SVRG gradients for solving different linear algebra problems.

gradient descent and thus SVRG will converge at this exponential rate [55, 16]. Below we list the gradients that enable us to solve different linear algebra problems: In each of the three cases listed above, we can recognize that if the average of all the gradients $g(w)$ is 0, then the corresponding linear algebra solution has been recovered.

While it may seem that we need to take three complete passes through $\{\mathbf{A}_i\}$ per SVRG epoch (due to the three terms in Equation 9), we can reduce this cost to two complete passes exploiting the fact that the gradients are linear in the matrix object, replacing $\mathbf{A}_i\mathbf{W} - \mathbf{A}_i\mathbf{W}_0$ with $\mathbf{A}_i(\mathbf{W} - \mathbf{W}_0)$ where appropriate. In all of the Sum structure experiments where we leverage SVRG, the x-axis measures the total number of passes through $\{\mathbf{A}_i\}_{i=1}^m$, two for each epoch for SVRG.

## C.2   Iterative methods

In Table 5 we list the different iterative methods (base cases) that we use for different linear algebraic operations as well as for different types of linear operators. As seen in Table 5, there are many alternatives to our base cases, however we opted for algorithms that are known to be performant, that are well-studied and that are popular amongst practitioners. A comprehensive explanation of our bases cases and their alternatives can be found in Golub and Loan [21] and Saad [44].

| Linear Algebra Op | Base Case | Alternatives |
|---|---|---|
| $\mathbf{A}\mathbf{x} = \mathbf{b}$ (non-symmetric) | GMRES | BiCGSTAB, LGMRES, QMR |
| $\mathbf{A}\mathbf{x} = \mathbf{b}$ (self-adjoint) | MINRES | GMRES |
| $\mathbf{A}\mathbf{x} = \mathbf{b}$ (PSD) | CG | GMRES |
| Eigs$(\mathbf{A})$ (non-symmetric) | Arnoldi | IRAM, Bi-Lanczos |
| Eigs$(\mathbf{A})$ (self-adjoint) | Lanczos | LOBPCG |
| $\mathbf{A}^+$ | CG | LSQR, LSMR |
| $\mathbf{A} = \mathbf{U}\boldsymbol{\Sigma}\mathbf{V}^*$ | Lanczos, rSVD | Jacobi-Davidson |
| $f(\mathbf{A})$ (self-adjoint) | SLQ | Arnoldi |

Table 5: **CoLA's base case iterative algorithm and some alternatives.** We now expand on the acronyms. GMRES: Generalized Minimum RESidual, BiCGSTAB: BiConjugate Gradient STABilized, QMR: Quasi-Minimal Residual, MINRES: MINimum RESidual, CG: Conjugate Gradients, IRAM: Implicitly Restarted Arnoldi Method, LOBPCG: Locally Optimal Block Preconditioned Conjugate Gradients, Bi-Lanczos: Bidiagonal Lanczos, CGS: Conjugate Gradient Squared, LSQR: Least squares QR, LSMR: Least squares Minimal Residual iteration, LGMRES: Least squares Generalized Minimum RESidual, rSVD: randomized Singular Value Decomposition, and SLQ: Stochastic Lanczos Quadrature.

## C.3   Lower precision linear algebra

The accumulation of round-off error is usually the breaking point of several numerical linear algebra (NLA) routines. As such, it is common to use precisions like `float64` or higher, especially when running these routines on a CPU. In contrast, in machine learning, lower precisions like `float32` or `float16` are ubiquitously used because more parameters and data can be fitted into the GPU memory (whose memory is usually much lower than CPUs) and because the MVMs can be done faster (the CUDA kernels are optimized for operations on these precisions). Additionally, the round-off error incurred on MVMs is not as detrimental when training machine learning models (as we are already

running noisy optimization algorithms) as when solving linear algebra problems (where round-off error can lead us to poor solutions). Thus, it is an active area of research in NLA to derive routines which utilize lower precisions than `float64` or that mix precisions in order to achieve better runtimes without a complete degradation of the quality of the solution.

In CoLA we take a two prong approach to deal with lower precisions in our NLA routines. First, we incorporate additional variants of well-known algorithms that propagate less round-off error at the expense of requiring more computation, as seen in Figure 7. Second, we integrate novel variants of algorithms that are designed to be used on lower precisions such as the CG modification found in Maddox et al. [32]. We now discuss the first approach.

As discussed in Section C.2, there are two algorithms that are key for eigendecompositions. The first is Arnoldi (applicable to any operator), and the second is Lanczos (for symmetric operators) — where actually Lanczos can be viewed as a simplified version of Arnoldi. Central to these algorithms is the use of an orthogonalization step which is well-known to be a source of numerical instability. One approach to aggressively ameliorate the propagation of round-off error during orthogonalization is to use Householder projectors, which is the strategy that we use in CoLA. Given a unitary vector $\mathbf{u}$, a Householder projector (or Householder reflector) is defined as the following operator $\mathbf{R} = \mathbf{I} - 2\mathbf{u}\mathbf{u}^*$. When applied to a vector $\mathbf{x}$ the result $\mathbf{R}\mathbf{x}$ is basically a reflection of $\mathbf{x}$ over the $\mathbf{u}^\intercal$ space. To easily visualize this, suppose that $\mathbf{x} \in \mathbb{R}^2$ and $\mathbf{u} = \mathbf{e}_1$. Hence,

$$\mathbf{R}\mathbf{x} = \begin{pmatrix} x_1 \\ x_2 \end{pmatrix} - 2 \begin{pmatrix} x_1 \\ 0 \end{pmatrix} = \begin{pmatrix} -x_1 \\ x_2 \end{pmatrix}$$

which is exactly the reflection of the vector across the axis generated by $\mathbf{e}_2$. Most notably, $\mathbf{R}$ is unitary $\mathbf{R}\mathbf{R}^* = \mathbf{I}$ which can be easily verified from the definition. Being unitary is crucial as under the usual round-off error model, applying $\mathbf{R}$ to another matrix $\mathbf{A}$ does not worsen the already accumulated error $\mathbf{E}$. Mathematically, $\|\mathbf{R}(\mathbf{A} + \mathbf{E}) - \mathbf{R}\mathbf{A}\| = \|\mathbf{R}\mathbf{E}\| = \|\mathbf{E}\|$, where the last equality results from basic properties of unitary matrices. We are going to use Arnoldi as an example of how Householder projectors are used during orthogonalization. In Figure 7 we have an example of two different variants of Arnoldi present in CoLA. The implementations are notably different and also it is easy to see how Algorithm 2 is more expensive than Algorithm 1. First, note that for Algorithm 2 we have two for loops (line 6 and line 8) whereas for Algorithm 1 we only have one (line 4-6). Worse, the two for loops in Algorithm 2 require more flops than the only for loop in Algorithm 1. Note that we do not always favor the more expensive but robust implementation of an algorithm as in some cases, like when running GMRES, the round-off error is not as impactful to the quality of the solution, and shorter runtimes are actually more desirable.

# D   Experimental Details

In this section we expand upon the details of all the experiments ran in the paper. Such details include the datasets that were used, the hyperparameters of different algorithms and the specific choices of algorithms used both for CoLA but also for the alternatives. We run each of the experiments 3 times and compute the mean dropping the first observation (as usually the first run contains some compiling time much is not too large). We do not display the standard deviation as those numbers are imperceptible for each experiment. In terms of hardware, the CPU experiments were run on an Intel(R) Core(TM) i5-9600K CPU @ 3.70GHz and the GPU experiments were run on a NVIDIA GeForce RTX 2080 Ti.

## D.1   Datasets

Below we enumerate the datasets that we used in the various applications. Most of the datasets are sourced from the University of California at Irvine's (UCI) Machine Learning Respository that can be found here: https://archive.ics.uci.edu/ml/datasets.php. Also, a community repo hosting these UCI benchmarks can be found here: https://github.com/treforevans/uci_datasets (we have no affiliation).

1. *Elevators*. This dataset is a modified version of the *Ailerons* dataset, where the goal is to to predict the control action on the ailerons of the aircraft. This UCI dataset consists of $N = 14$K observations and has $D = 18$ dimensions.

| **Algorithm 1** Arnoldi iteration | **Algorithm 2** Householder Arnoldi iteration |
|---|---|
| 1: **Inputs: A**, $\mathbf{q}_0 = \boldsymbol{\nu}_0 / \|\boldsymbol{\nu}_0\|$ where possibly $\boldsymbol{\nu}_0 \sim \mathcal{N}(\mathbf{0}, \mathbf{I})$, maximum number of iterations $T$ and tolerance $\epsilon \in (0,1)$. | 1: **Inputs: A**, $\boldsymbol{\nu}_0 \neq \mathbf{0}$ where possibly $\boldsymbol{\nu}_0 \sim \mathcal{N}(\mathbf{0}, \mathbf{I})$, and maximum number of iterations $T$. |
| 2: **for** $j = 0$ to $T-1$ **do** | 2: **for** $j = 0$ to $T$ **do** |
| 3:    $\boldsymbol{\nu}_{j+1} \leftarrow \mathbf{A}\mathbf{q}_j$ | 3:    $\mathbf{u}_j = \texttt{GET\_HOUSEHOLDER\_VEC}(\boldsymbol{\nu}_j, j)$ |
| 4:    **for** $i = 0$ to $j$ **do** | 4:    $\mathbf{R}_j = \mathbf{I} - 2\mathbf{u}_j \mathbf{u}_j^*$ |
| 5:      $h_{i,j} = \mathbf{q}_i^*(\mathbf{A}\mathbf{q}_j)$ | 5:    $\mathbf{h}_j = \mathbf{R}_j \boldsymbol{\nu}_j$ |
| 6:      $\boldsymbol{\nu}_{j+1} \leftarrow \boldsymbol{\nu}_{j+1} - h_{i,j}\mathbf{q}_i$ | 6:    $\mathbf{q}_j = \mathbf{R}_0 \cdots \mathbf{R}_j \mathbf{e}_{j+1}$ |
| 7:    **end for** | 7:    **if** $j < T$ **then** |
| 8:    $h_{j+1,j} = \|\boldsymbol{\nu}_{j+1}\|$ | 8:      $\boldsymbol{\nu}_{j+1} = \mathbf{R}_j \cdots \mathbf{R}_0 (\mathbf{A}\mathbf{q}_j)$ |
| 9:    **if** $h_{j+1,j} < \epsilon$ **then** | 9:    **end if** |
| 10:      **stop** | 10: **end for** |
| 11:    **else** | 11: **return** $\mathbf{H}, \mathbf{Q} = (\mathbf{q}_0 | \dots | \mathbf{q}_T)$ |
| 12:      $\mathbf{q}_{j+1} = \boldsymbol{\nu}_{j+1}/h_{j+1,j}$ | 12: **function** $\texttt{GET\_HOUSEHOLDER\_VEC}(\mathbf{w}, k)$ |
| 13:    **end if** | 13:    $u_i = 0$ for $i < k$ and $u_i = w_i$ for $i > k$. |
| 14: **end for** | 14:    $u_k = w_k - \|\mathbf{w}\|$ |
| 15: **return** $\mathbf{H}, \mathbf{Q} = (\mathbf{q}_0 | \dots | \mathbf{q}_{T-1} | \mathbf{q}_T)$ | 15:    **return u** |
| | 16: **end function** |

Figure 7: Different versions of the same algorithm, but the Householder variant being more numerically robust.

2. *Kin40K*. The full name of this UCI dataset is *Statlog (Shuttle) Data Set*. This dataset contains information about NASA shuttle flights and we used a subset that consists of $N = 40$K observations and has $D = 8$ dimensions.

3. *Buzz*. The full name of this UCI dataset is *Buzz in social media*. This dataset consists of examples of buzz events from Twitter and Tom's Hardware. We used a subset consisting of $N = 430$K observations and has $D = 77$ dimensions.

4. *Song*. The full name of this UCI dataset is *YearPredictionMSD*. This dataset consists of $N = 386.5$K observations and it has $D = 90$ audio features such as 12 timbre average features and 78 timbre covariance features.

5. *cit-HepPh*. This dataset is based on arXiv's HEP-PH (high energy physics phenomenology) citation graph and can be found here: https://snap.stanford.edu/data/cit-HepPh.html. The dataset covers all the citations from January 1993 to April 2003 of $|V| = 34,549$ papers, ultimately containing $|E| = 421,578$ directed edges. The notion of relationship that we used in our spectral clustering experiment creates a connection between two papers when at least one cites another (undirected symmetric graph). Therefore the dataset that we used has the same number of nodes but instead $|E| = 841,798$ undirected edges.

### D.2 Compositional experiments

This section pertains to the experiments of Section 3.2 displayed in Figure 1. We now elaborate on each of Figure 1's panels.

(a) The multi-task GP problem exploits the structure of the following Kronecker operator $\mathbf{K}_T \otimes \mathbf{K}_X$, where $\mathbf{K}_T$ is a kernel matrix containing the correlation between the tasks and $\mathbf{K}_X$ is a RBF kernel on the data. For this experiment, we used a synthetic Gaussian dataset where the train data $\mathbf{x}_i \sim \mathcal{N}(\mathbf{0}, \mathbf{I}_D)$ which has dimension $D = 33$, $N = 1$K and we used $T = 11$ tasks (where the tasks basically set the size of $\mathbf{K}_T$). We used conjugate gradients (CG) as the iterative method, where we set the hyperparameters to a tolerance of $10^{-6}$ and to a maximum number of iterations to 1K. We used the exact same hyperparameters for CoLA.

(b) For the bi-poisson problem we set up the maximum grid to be $N = 1000^2$. Since this PDE problem involves solving a symmetric linear system, we used CG as the iterative method with a tolerance of $10^{-11}$ and a maximum number of iterations of 10K. The previous

parameters also apply for CoLA. We note that PDE problems are usually solved to higher tolerances as the numerical error compounds as we advance the PDE.

(c) For the EMLP experiment we consider solving the equivariance constraints to find the equivariant linear layers of a graph neural network with 5 nodes. To solve this problem, we need to find the nullspace of a large structured constraint matrix. We use the uniformly channel heuristic from [16] which distributes the $N$ channels across tensors of different orders. We consider our approach which exploits the block diagonal structure, separating the nullspaces into blocks, as opposed to the direct iterative approach exploiting only the fast MVMs of the constraint matrix. We use a tolerance of $10^{-5}$.

## D.3 Sum structure experiments

This section pertains to the experiments of Section 3.3 contained in Figure 2. We now elaborate on each of Figure 2's panels.

(a) In this experiment we computed the first principal component of the *Buzz* dataset. For the iterative method we used power iteration with a maximum number of iterations of 300 and a stop tolerance of $10^{-7}$. CoLA used SVRG also with the same stop tolerance and maximum number of iterations. Additionally, we set SVRG's batch size to 10K and the learning rate to 0.0008. We note that a single power iteration roughly contains $43/2 = 21.5$ times more MVMs than a single iteration of SVRG. In this particular case, the length of the sum is given by the number of observations and therefore SVRG uses $430/10 = 43$ times less elements per iteration, where 10 comes from the 10K batch size. Finally, the 2 is explained by noting that SVRG incurs in a full sum update on every epoch.

(b) In this experiment we trained a GP by estimating the covariance RBF kernel with $J = 1K$ random Fourier features (RFFs). The hyperparameters for the RBF kernel are the following: length scale ($\ell = 0.1$), output scale ($a = 1$) and likelihood noise ($\sigma^2 = 0.1$). Moreover, we used CG as the iterative solver with a tolerance of $10^{-8}$ and 100 as the maximum number of iterations (the convergence took much less iterations than the max). For SVRG we used the same tolerance but set the maximum number of iterations to 10K, a batch size of 100 and learning rate of 0.004. We note that a single CG iteration roughly contains $10/2 = 5$ times more MVMs than a single iteration of SVRG. In this particular case, the length of the sum is given by the number of RFFs and therefore SVRG uses $1000/100 = 10$ times less elements per iteration, where 100 comes from the batch size.

(c) In this experiment we implemented the Neural-IVP method from Finzi et al. [17]. We consider the time evolution of a wave equation in two spatial dimensions. At each integrator step, a linear system $\mathbf{M}(\theta)\dot{\theta} = F(\theta)$ must be solved to find $\dot{\theta}$, for a $d = 12K \times 12K$ dimensional matrix. While Finzi et al. [17] use conjugate gradients to solve the linear system, we demonstrate the advantages of using SVRG, as $\mathbf{M}(\theta) = \frac{1}{m}\sum_{i=1}^{m} M_i(\theta)$ is a sum over the evaluation at $m = 50K$ distinct sample locations within the domain. In this experiment we use a batch size of 500 for SVRG, and employ rank 250 randomized Nyström preconditioning for both SVRG and the iterative CG baseline.

## D.4 Hardware speed-up comparisons

This section pertains to the experiments of Figure 3. For all these experiments we computed the runtime reduction as a fraction between the time that it takes CoLA to run some linear algebra operation and PyTorch using the same hardware. As an example, assume that PyTorch takes 200 seconds to compute a solve using a CPU and 100 seconds to compute the same solve but now using a GPU. Moreover, assume that CoLA's iterative algorithm takes 100 seconds to compute the same solve on a CPU and 40 seconds on a GPU. Thus, the runtime reduction would be $100/200 = 0.5\%$ for the CPU column whereas $40/100 = 0.4$ for the GPU column.

1. **Solves**. In this experiment we calculated the % runtime reduction when running `torch.linalg.solve` on the Trefethen $N = 20K$ matrix market sparse operator. In this experiment, CG was run with a tolerance of $10^{-11}$ and a maximum number of iterations equal to the operator size.

2. **Eigenvalue estimation**. In this experiment we calculated the % runtime reduction when running `torch.linalg.eigh` on the mhd4800b $N = 4.8K$ matrix market sparse operator. In this experiment, Lanczos was run with a tolerance of $10^{-9}$ and a maximum number of iterations equal to 100.

3. **Log determinant computation**. In this experiment we calculated the % runtime reduction when running `torch.linalg.logdet` on the bcsstk18 $N = 11.9K$ matrix market sparse operator. In this experiment, the stochastic Lanczos quadrature was run using 30 Lanczos probe estimates and 25 samples.

### D.5   Applications

This section pertains to the experiments of Section 4 displayed in Figure 4. We now elaborate on each of Figure 4's panels.

(a) In this experiment we compute 5, 10 and 20 PCA components for the *Buzz* dataset. We compared against `sklearn` which uses the Lanczos algorithm through the fast Fortran-based `ARPACK` numerical library. In this case, CoLA uses randomized SVD [35] with a rank 3000 approximation.

(b) In this experiment we fit a Ridge regression on the *Song* dataset with a regularization coefficient set to 0.1. We compared against `sklearn` using their fastest least-square solver `lsqr` with a tolerance of $10^{-4}$. In this case, CoLA uses CG with the same tolerance and with a maximum number of iterations set to 1K. Additionally, we ran CoLA using CPU and GPU whereas we used only CPU for `sklearn` as it has no GPU support. We observe how in the arguably most popular ML method, CoLA is able to beat a leading package such as `sklearn`.

(c) In this experiment we fit a GP with a RBF kernel on two datasets: *Elevators* and *Kin40K*. We only used up to 20K observations from *Kin40K* as that was the maximum number of observations that would fit the GPU memory without needing to partition the MVMs. We compare against `GPyTorch` which uses CG and stochastic Lanczos quadrature (SLQ) to compute and optimize the negative log-marginal likelihood (loss function). Both experiments were run on a GPU for 100 iterations using Adam as an optimizer with learning rate of 0.1 with the default values of $\beta_1 = 0.9$ and $\beta_2 = 0.999$. Additionally, for both GPyTorch and CoLA, the CG tolerance was set to $10^{-4}$ with a maximum number of CG iterations of 250 and 20 probes were used for SLQ. Note that both CoLA and GPyTorch have similar throughputs, for example GPyTorch runs a 100 iterations on *Elevators* on 43 seconds whereas CoLA runs a 100 iterations on 49 seconds. When training a GP, we solve a block of 11 linear systems (1 based on $\mathbf{y}$ and 10 based on random probes) where one key difference is that the CG solver for GPyTorch has a stopping criteria based on the convergence of the mean solves whereas CoLA has a stopping criteria based on the convergence of all the solves.

(d) In this experiment we run spectral clustering on the *cit-HepPh* dataset using an embedding size of 8 and also 8 clusters for k-means (with only 1 run of k-means after estimating the embeddings). We compare against `sklearn` using two different solvers, one based on Lanczos iterations using `ARPACK` and another using an Algebraic Multi-Grid solver `AMG`. In this case, CoLA also uses Lanczos iterations with a default tolerance of $10^{-6}$. We see how `sklearn`'s `AMG` solver runs faster than CoLA's but this is mostly the algorithmic constants as they have similar asymptotical behavior (similar slopes).

(e) In this experiment we solve the Schrödinger equation to find the energy levels of the hydrogen atom on a 3-dimensional finite difference grid with up to $N = 5K$ points. In order to handle the infinite spatial extent, we compactify the domain by applying the arctan function. Under this change of coordinates, the Laplacian has a different form, and hence the matrix forming the discretized Hamiltonian is no longer symmetric. We compare against `SciPy`'s Arnoldi implementation with 20 iterations where CoLA also uses Arnoldi with the same number of iterations. Surprisingly, CoLA's JAX jitted code has a competitive runtime when compare to `SciPy`'s runtime using `ARPACK`.

(f) In this experiment we solve a minimal surface problem on a grid of maximum size of $N = 100^2$ points. To solve this problem we have to run Netwon-Rhapson where each

inner step involves a linear solve of an non-symmetric operator. We compare against SciPy's GMRES implementation as well as JAX's integrated version of SciPy. The main difference between the two is that SciPy calls the fast and highly-optimized ARPACK library whereas SciPy (JAX) has its only Python implementation of GMRES which only uses JAX's primitives (equally as it is done in CoLA). The tolerance for this experiment was 5e-3. We see how CoLA's GMRES implementation is competitive with SciPy (JAX) but it still does not beat ARPACK mostly due to the faster runtime of using a lower level GMRES implementation.

