## Appendix Outline

491 This Appendix is organized as follows:

492 • In Appendix A we describe various dispatch rules including the base rules, the composition
493 rules and rules derived from other rules.

494 • In Appendix B we provide an extended discussion of several noteworthy features of CoLA,
495 such as doubly stochastic estimators and memory-efficient autograd implementation.

496 • In Appendix C we include pseudo-code on various of the iterative methods incorporated in
497 CoLA and discuss modifications to improve lower precision performance.

498 • In Appendix D we expand on the details of the experiments in the main text.

499 • In Appendix E we show some code examples of how the dispatch rules are implemented in
500 CoLA.

## A   Dispatch Rules

502 We now present the linear algebra identities that we use to exploit structure in CoLA.

### A.1   Core Functions

### A.1.1   Inverses

505 We incorporate several identities for the compositional operators: product, Kronecker product, block
506 diagonal and sum. For product we have $(\mathbf{AB})^{-1} = (\mathbf{B}^{-1}\mathbf{A}^{-1})$ and for Kronecker product we have
507 $(\mathbf{A} \otimes \mathbf{B})^{-1} = \mathbf{A}^{-1} \otimes \mathbf{B}^{-1}$. In terms of block compositions we have the following identities:

$$\begin{bmatrix} \mathbf{A} & \mathbf{0} \\ \mathbf{0} & \mathbf{D} \end{bmatrix}^{-1} = \begin{bmatrix} \mathbf{A}^{-1} & \mathbf{0} \\ \mathbf{0} & \mathbf{D}^{-1} \end{bmatrix} \quad \text{and} \quad \begin{bmatrix} \mathbf{A} & \mathbf{B} \\ \mathbf{0} & \mathbf{D} \end{bmatrix}^{-1} = \begin{bmatrix} \mathbf{A}^{-1} & -\mathbf{A}^{-1}\mathbf{B}\mathbf{D}^{-1} \\ \mathbf{0} & \mathbf{D}^{-1} \end{bmatrix}$$

$$\begin{bmatrix} \mathbf{A} & \mathbf{B} \\ \mathbf{C} & \mathbf{D} \end{bmatrix}^{-1} = \begin{bmatrix} \mathbf{I} & -\mathbf{A}^{-1}\mathbf{B} \\ \mathbf{0} & \mathbf{I} \end{bmatrix} \begin{bmatrix} \mathbf{A} & \mathbf{0} \\ \mathbf{0} & \mathbf{D} - \mathbf{C}\mathbf{A}^{-1}\mathbf{B} \end{bmatrix}^{-1} \begin{bmatrix} \mathbf{I} & \mathbf{0} \\ -\mathbf{C}\mathbf{A}^{-1} & \mathbf{I} \end{bmatrix}$$

509 Finally, for sum we have the Woodbury identity and its variants. Namely, for Woodbury we have

$$(\mathbf{A} + \mathbf{U}\mathbf{B}\mathbf{V})^{-1} = \mathbf{A}^{-1} - \mathbf{A}^{-1}\mathbf{U}\left(\mathbf{B}^{-1} + \mathbf{V}\mathbf{A}^{-1}\mathbf{U}\right)^{-1}\mathbf{V}\mathbf{A}^{-1},$$

510 the Kailath variant where

$$(\mathbf{A} + \mathbf{B}\mathbf{C})^{-1} = \mathbf{A}^{-1} - \mathbf{A}^{-1}\mathbf{B}\left(\mathbf{I} + \mathbf{C}\mathbf{A}^{-1}\mathbf{B}\right)\mathbf{C}\mathbf{A}^{-1}$$

511 and the rank one update via the Sherman-Morrison formula

$$(\mathbf{A} + \mathbf{b}\mathbf{c}^{\mathsf{T}})^{-1} = \mathbf{A}^{-1} - \frac{1}{1 + \mathbf{c}^{\mathsf{T}}\mathbf{A}\mathbf{b}}\mathbf{A}^{-1}\mathbf{b}\mathbf{c}^{\mathsf{T}}\mathbf{A}^{-1}.$$

512 Besides the compositional operators, we have some rules for some special operators. For example,
513 for $\mathbf{A} = \texttt{Diag}\,(\mathbf{a})$ we have $\mathbf{A}^{-1} = \texttt{Diag}\,(\mathbf{a}^{-1})$. Also, if $\mathbf{Q}$ is unitary then $\mathbf{Q}^{-1} = \mathbf{Q}^*$ or if $\mathbf{Q}$ is
514 orthonormal then $\mathbf{Q}^{-1} = \mathbf{Q}^{\mathsf{T}}$. In Appendix E we show how these dispatch rules are implemented in
515 `Python`.

### A.1.2   Eigendecomposition

517 We now assume that the matrices in this section are diagonalizable. That is, $\texttt{Eigs}\,(\mathbf{A}) = \mathbf{\Lambda_A}, \mathbf{V_A}$,
518 where $\mathbf{A} = \mathbf{V_A}\mathbf{\Lambda_A}\mathbf{V_A}^{-1}$. In terms of the compositional operators, there is not a general rule for
519 product or sum. However, for the Kronecker product we have $\texttt{Eigs}(\mathbf{A} \otimes \mathbf{B}) = \mathbf{\Lambda_A} \otimes \mathbf{\Lambda_B},\ \mathbf{V_A} \otimes \mathbf{V_B}$
520 and for the Kronecker sum we have $\texttt{Eigs}(\mathbf{A} \oplus \mathbf{B}) = \mathbf{\Lambda_A} \oplus \mathbf{\Lambda_B},\ \mathbf{V_A} \otimes \mathbf{V_B}$. Finally, for block
521 diagonal we have

$$\texttt{Eigs}\left(\begin{bmatrix} \mathbf{A} & \mathbf{0} \\ \mathbf{0} & \mathbf{D} \end{bmatrix}\right) = \begin{bmatrix} \mathbf{\Lambda_A} & \mathbf{0} \\ \mathbf{0} & \mathbf{\Lambda_D} \end{bmatrix}, \begin{bmatrix} \mathbf{V_A} & \mathbf{0} \\ \mathbf{0} & \mathbf{V_D} \end{bmatrix}.$$

### A.1.3 Diagonal

As a base case, if we need to compute $\text{Diag}\,(\mathbf{A})$ for a general matrix $\mathbf{A}$ we may compute each diagonal element by $\mathbf{e}_i^\mathsf{T}\mathbf{A}\mathbf{e}_i$. Additionally, if $\mathbf{A}$ is large enough we switch to randomized estimation $\text{Diag}(\mathbf{A}) \approx (\mathbf{Z} \odot \mathbf{A}\mathbf{Z})\mathbf{1}/N$ with $\mathbf{Z} \sim \mathcal{N}(0,1)^{d\times N}$ where $N$ is the number of samples used to approximate the diagonal. In terms of compositional operators, we have that for sum $\text{Diag}\,(\mathbf{A}+\mathbf{B}) = \text{Diag}\,(\mathbf{A}) + \text{Diag}\,(\mathbf{B})$. For Kronecker product we have $\text{Diag}(\mathbf{A}\otimes\mathbf{B}) = \text{vec}\big(\text{Diag}(\mathbf{A})\text{Diag}(\mathbf{B})^\mathsf{T}\big)$ and for Kronecker sum $\text{Diag}(\mathbf{A}\oplus\mathbf{B}) = \text{vec}\big(\text{Diag}\,(\mathbf{A})\,\mathbf{1}^\mathsf{T} + \mathbf{1}\text{Diag}\,(\mathbf{B})^\mathsf{T}\big)$. Finally, for block composition we have

$$\text{Diag}\left(\begin{bmatrix} \mathbf{A} & \mathbf{B} \\ \mathbf{C} & \mathbf{D} \end{bmatrix}\right) = [\text{Diag}(\mathbf{A}), \text{Diag}(\mathbf{D})].$$

### A.1.4 Transpose / Adjoint

As explained in Section 3.1, as a base case we have an automatic procedure to compute the transpose or adjoint of any operator $\mathbf{A}$ via autodiff. However, we also incorporate the following rules. For sum we have $(\mathbf{A}+\mathbf{B})^* = \mathbf{A}^* + \mathbf{B}^*$ and $(\mathbf{A}+\mathbf{B})^\mathsf{T} = \mathbf{A}^\mathsf{T} + \mathbf{B}^\mathsf{T}$. For product we have $(\mathbf{A}\mathbf{B})^* = \mathbf{B}^*\mathbf{A}^*$ and $(\mathbf{A}\mathbf{B})^\mathsf{T} = \mathbf{B}^\mathsf{T}\mathbf{A}^\mathsf{T}$. For Kronecker product we have $(\mathbf{A}\otimes\mathbf{B})^* = \mathbf{A}^*\otimes\mathbf{B}^*$ and $(\mathbf{A}\otimes\mathbf{B})^\mathsf{T} = \mathbf{A}^\mathsf{T}\otimes\mathbf{B}^\mathsf{T}$. For the Kronecker sum we have $(\mathbf{A}\oplus\mathbf{B})^* = \mathbf{A}^*\oplus\mathbf{B}^*$ and $(\mathbf{A}\oplus\mathbf{B})^\mathsf{T} = \mathbf{A}^\mathsf{T}\oplus\mathbf{B}^\mathsf{T}$. In terms of block composition we have

$$\left(\begin{bmatrix} \mathbf{A} & \mathbf{B} \\ \mathbf{C} & \mathbf{D} \end{bmatrix}\right)^* = \begin{bmatrix} \mathbf{A}^* & \mathbf{C}^* \\ \mathbf{B}^* & \mathbf{D}^* \end{bmatrix} \quad\text{and}\quad \left(\begin{bmatrix} \mathbf{A} & \mathbf{B} \\ \mathbf{C} & \mathbf{D} \end{bmatrix}\right)^\mathsf{T} = \begin{bmatrix} \mathbf{A}^\mathsf{T} & \mathbf{C}^\mathsf{T} \\ \mathbf{B}^\mathsf{T} & \mathbf{D}^\mathsf{T} \end{bmatrix}.$$

Finally for the annotated operators we have the following rules. $\mathbf{A}^* = \mathbf{A}$ if $\mathbf{A}$ is self-adjoint and $\mathbf{A}^\mathsf{T} = \mathbf{A}$ if $\mathbf{A}$ is symmetric.

### A.1.5 Pseudo-inverse

As a base case, if we need to compute $\mathbf{A}^+$, we may use $\text{SVD}\,(\mathbf{A}) = \mathbf{U}, \mathbf{\Sigma}, \mathbf{V}$ and therefore set $\mathbf{A}^+ = \mathbf{U}\mathbf{\Sigma}^+\mathbf{V}^*$, where $\mathbf{\Sigma}^+$ inverts the nonzero diagonal scalars. If the size of $\mathbf{A}$ is too large, then we may use randomized SVD. Yet, it is uncommon to simply want $\mathbf{A}^+$, usually we want to solve a least-squares problem and therefore we can use solvers that are not as expensive to run as SVD. For the compositional operators we have the following identities. For product $(\mathbf{A}\mathbf{B})^+ = \left(\mathbf{A}^+\mathbf{A}\mathbf{B}\right)^+ \left(\mathbf{A}\mathbf{B}\mathbf{B}^+\right)^+$ and for Kronecker product we have $(\mathbf{A}\otimes\mathbf{B})^+ = \mathbf{A}^+\otimes\mathbf{B}^+$. For block diagonal we have

$$\left(\begin{bmatrix} \mathbf{A} & \mathbf{0} \\ \mathbf{0} & \mathbf{D} \end{bmatrix}\right)^+ = \begin{bmatrix} \mathbf{A}^+ & \mathbf{0} \\ \mathbf{0} & \mathbf{D}^+ \end{bmatrix}.$$

Finally, we have some identities that are mathematically trivial but that are necessary when recursively exploiting structure as that would save computation. For example, if $\mathbf{Q}$ is unitary we know that $\mathbf{Q}^+ = \mathbf{Q}$ and similarly when $\mathbf{Q}$ is orthonormal. If $\mathbf{A}$ is self-adjoint, then $\mathbf{A}^+ = \mathbf{A}^{-1}$ and also if it is symmetric and PSD.

## A.2 Derived Functions

Interestingly, the previous core functions allow us to derive multiple rules from the previous ones. To illustrate, we have that $\text{Tr}\,(\mathbf{A}) = \sum_i \text{Diag}\,(\mathbf{A})_i$. Additionally, if $\mathbf{A}$ is PSD we have that $f\,(\mathbf{A}) = \mathbf{V}_\mathbf{A} f\,(\mathbf{\Lambda}_\mathbf{A})\mathbf{V}_\mathbf{A}^{-1}$ and if $\mathbf{A}$ is both symmetric and PSD then $f\,(\mathbf{A}) = \mathbf{V}_\mathbf{A} f\,(\mathbf{\Lambda}_\mathbf{A})\mathbf{V}_\mathbf{A}^\mathsf{T}$. where in both cases we used $\text{Eigs}\,(\mathbf{A}) = \mathbf{\Lambda}_\mathbf{A}, \mathbf{V}_\mathbf{A}$. Some example functions for PSD matrices are $\text{Sqrt}\,(\mathbf{A}) = \mathbf{V}_\mathbf{A}\mathbf{\Lambda}_\mathbf{A}^{1/2}\mathbf{V}_\mathbf{A}^{-1}$ or $\text{Log}\,(\mathbf{A}) = \mathbf{V}_\mathbf{A}\log\mathbf{\Lambda}_\mathbf{A}\mathbf{V}_\mathbf{A}^{-1}$. Which also this rules allow us to define $\text{LogDet}\,(\mathbf{A}) = \text{Tr}\,(\text{Log}\,(\mathbf{A}))$.

## A.3 Other matrix identities

We emphasize that there are a myriad more matrix identities that we do not intentionally include such as $\text{Tr}(\mathbf{A}+\mathbf{B}) = \text{Tr}(\mathbf{A}) + \text{Tr}(\mathbf{B})$ or $\text{Tr}(\mathbf{A}\mathbf{B}) = \text{Tr}(\mathbf{B}\mathbf{A})$ when $\mathbf{A}$ and $\mathbf{B}$ are squared. These

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

$$\mathbf{v}^{\intercal} \frac{\partial f}{\partial \mathbf{A}} = \mathtt{unflatten}\big[\mathtt{VJP}\big(\theta \mapsto \mathtt{unflatten}(\theta)\mathbf{A}^{-1}\mathbf{b}, \theta, \mathbf{A}^{-1}\mathbf{v}\big)\big] \tag{8}$$

for $\mathbf{v}^{\intercal} \frac{\partial f}{\partial \theta} = \mathbf{v}^{\intercal}(\mathbf{A}^{-1})^{\intercal}(\partial_{\theta}\mathbf{A}_{\theta})\mathbf{A}^{-1}\mathbf{b}$, and we will adopt this notation below for brevity. Doing so gives a memory cost which is constant in the number of solver iterations, and proportional to the memory used in the forward pass. Below we list the autograd rules for some of the iterative routines that we implement in CoLA with their $\mathtt{VJP}$ definitions.

1. $\mathbf{y} = \mathtt{Solve}(\mathbf{A}, \mathbf{b})$ :   $\mathbf{w}^{\intercal} \frac{\partial \mathbf{y}}{\partial \boldsymbol{\theta}} = -(\mathbf{A}^{-1}\mathbf{w})^{\intercal}(\partial_{\boldsymbol{\theta}}\mathbf{A}_{\boldsymbol{\theta}})(\mathbf{A}^{-1}\mathbf{b})$

2. $\boldsymbol{\lambda}, \mathbf{V} = \mathtt{Eigs}(\mathbf{A})$ :   $\mathbf{w}^{\intercal} \frac{\partial \boldsymbol{\lambda}}{\partial \boldsymbol{\theta}} = \mathbf{w}^{\intercal}\mathtt{Diag}\big(\mathbf{V}^{\intercal}(\partial_{\boldsymbol{\theta}}\mathbf{A}_{\boldsymbol{\theta}})\mathbf{V}\big)$

3. $\boldsymbol{\lambda}, \mathbf{V} = \mathtt{Eigs}(\mathbf{A})$ :   $\mathbf{w}^{\intercal} \frac{\partial \mathbf{v}_i}{\partial \boldsymbol{\theta}} = \mathbf{w}^{\intercal}(\lambda_i\mathbf{I} - \mathbf{A})^{+}\partial_{\boldsymbol{\theta}}\mathbf{A}_{\boldsymbol{\theta}}\mathbf{v}_i$

4. $y = \log|\mathbf{A}|$ :   $\frac{\partial y}{\partial \boldsymbol{\theta}} = \mathtt{Tr}\big(\mathbf{A}^{-1}\partial_{\boldsymbol{\theta}}\mathbf{A}_{\boldsymbol{\theta}}\big)$

5. $\mathbf{y} = \mathtt{Diag}(\mathbf{A})$ :   $\mathbf{w}^{\intercal} \frac{\partial \mathbf{y}}{\partial \boldsymbol{\theta}} = \mathbf{w}^{\intercal}\mathtt{Diag}\big(\partial_{\boldsymbol{\theta}}\mathbf{A}_{\boldsymbol{\theta}}\big)$

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

| Linear Algebra Op | Base Case | Alternatives |
|---|---|---|
| $\mathbf{Ax} = \mathbf{b}$ (asymmetric) | GMRES | BiCGSTAB, CR, QMR |
| $\mathbf{Ax} = \mathbf{b}$ (self-adjoint) | MINRES | GMRES |
| $\mathbf{Ax} = \mathbf{b}$ (PSD) | CG | GMRES |
| `Eigs`$(\mathbf{A})$ (asymmetric) | Arnoldi | IRAM, QR algorithm |
| `Eigs`$(\mathbf{A})$ (self-adjoint) | Lanczos | LOBPCG, Rayleigh-Ritz, Bi-Lanczos |
| $\mathbf{A}^{+}$ | CG | CGS, LSQR, LGMRES |
| $\mathbf{A} = \mathbf{U\Sigma V}^{*}$ | Lanczos, rSVD | Jacobi-Davidson |
| $f(\mathbf{A})$ (self-adjoint) | SLQ | SVD, Rational Krylov Subspaces |

Table 5: **CoLA's base case iterative algorithm and some alternatives.** We now expand on the acronyms. GMRES: Generalized Minimum RESidual, BiCGSTAB: BiConjugate Gradient STABilized, CR: Conjugate Residuals, QMR: Quasi-Minimal Residual, MINRES: MINimum RESidual, CG: Conjugate Gradients, IRAM: Implicitly Restarted Arnoldi Method, LOBPCG: Locally Optimal Block Preconditioned Conjugate Gradients, Bi-Lanczos: Bidiagonal Lanczos, CGS: Conjugate Gradient Squared, LSQR: Least-Squares QR, LGMRES: Least-sqaures Generalized Minimum RESidual, SVD: Singular Value Decomposition, rSVD: randomized Singular Value Decomposition, and SLQ: Stochastic Lanczos Quadrature.

expense of requiring more computation, as seen in Figure 5. Second, we integrate novel variants of algorithms that are designed to be used on lower precisions such as the CG modification found in Maddox et al. [28]. We now discuss the first approach.

As discussed in Section C.2, there are two algorithms that are key for eigendecompositions. The first is Arnoldi (applicable to any operator), and the second is Lanczos (for symmetric operators) — where actually Lanczos can be viewed as a simplified version of Arnoldi. Central to these algorithms is the use of an orthogonalization step which is well-known to be a source of numerical instability. One approach to aggressively ameliorate the propagation of round-off error during orthogonalization is to use Householder projectors, which is the strategy that we use in CoLA. Given a unitary vector $\mathbf{u}$, a Householder projector (or Householder reflector) is defined as the following operator $\mathbf{R} = \mathbf{I} - 2\mathbf{uu}^{*}$. When applied to a vector $\mathbf{x}$ the result $\mathbf{Rx}$ is basically a reflection of $\mathbf{x}$ over the $\mathbf{u}^{\mathsf{T}}$ space. To easily visualize this, suppose that $\mathbf{x} \in \mathbb{R}^2$ and $\mathbf{u} = \mathbf{e}_1$. Hence,

$$\mathbf{Rx} = \begin{pmatrix} x_1 \\ x_2 \end{pmatrix} - 2 \begin{pmatrix} x_1 \\ 0 \end{pmatrix} = \begin{pmatrix} -x_1 \\ x_2 \end{pmatrix}$$

which is exactly the reflection of the vector across the axis generated by $\mathbf{e}_2$. Most notably, $\mathbf{R}$ is unitary $\mathbf{RR}^{*} = \mathbf{I}$ which can be easily verified from the definition. Being unitary is crucial as under the usual round-off error model, applying $\mathbf{R}$ to another matrix $\mathbf{A}$ does not worsen the already accumulated error $\mathbf{E}$. Mathematically, $\|\mathbf{R}(\mathbf{A} + \mathbf{E}) - \mathbf{RA}\| = \|\mathbf{RE}\| = \|\mathbf{E}\|$, where the last equality results from basic properties of unitary matrices. We are going to use Arnoldi as an example of how Householder projectors are used during orthogonalization. In Figure 5 we have an example of two different variants of Arnoldi present in CoLA. The implementations are notably different and also it is easy to see how Algorithm 2 is more expensive than Algorithm 1. First, note that for Algorithm 2 we have two for loops (line 6 and line 8) whereas for Algorithm 1 we only have one (line 4-6). Worse, the two for loops in Algorithm 2 require more flops than the only for loop in Algorithm 1. Note that we do not always favor the more expensive but robust implementation of an algorithm as in some cases, like when running GMRES, the round-off error is not as impactful to the quality of the solution, and shorter runtimes are actually more desirable.

# D   Experimental Details

In this section we expand upon the details of all the experiments ran in the paper. Such details include the datasets that were used, the hyperparameters of different algorithms and the specific choices

| **Algorithm 1** Arnoldi iteration | **Algorithm 2** Householder Arnoldi iteration |
|---|---|
| 1: **Inputs: A**, $\mathbf{q}_0 = \boldsymbol{\nu}_0/\|\boldsymbol{\nu}_0\|$ where possibly $\boldsymbol{\nu}_0 \sim \mathcal{N}(\mathbf{0}, \mathbf{I})$, maximum number of iterations $T$ and tolerance $\epsilon \in (0,1)$. | 1: **Inputs: A**, $\boldsymbol{\nu}_0 \neq \mathbf{0}$ where possibly $\boldsymbol{\nu}_0 \sim \mathcal{N}(\mathbf{0}, \mathbf{I})$, and maximum number of iterations $T$. |
| 2: **for** $j = 0$ to $T-1$ **do** | 2: **for** $j = 0$ to $T$ **do** |
| 3: $\quad \boldsymbol{\nu}_{j+1} \leftarrow \mathbf{A}\mathbf{q}_j$ | 3: $\quad \mathbf{u}_j = \texttt{GET\_HOUSEHOLDER\_VEC}(\boldsymbol{\nu}_j, j)$ |
| 4: $\quad$ **for** $i = 0$ to $j$ **do** | 4: $\quad \mathbf{R}_j = \mathbf{I} - 2\mathbf{u}_j\mathbf{u}_j^*$ |
| 5: $\qquad h_{i,j} = \mathbf{q}_i^*(\mathbf{A}\mathbf{q}_j)$ | 5: $\quad \mathbf{h}_j = \mathbf{R}_j\boldsymbol{\nu}_j$ |
| 6: $\qquad \boldsymbol{\nu}_{j+1} \leftarrow \boldsymbol{\nu}_{j+1} - h_{i,j}\mathbf{q}_i$ | 6: $\quad \mathbf{q}_j = \mathbf{R}_0 \cdots \mathbf{R}_j\mathbf{e}_{j+1}$ |
| 7: $\quad$ **end for** | 7: $\quad$ **if** $j < T$ **then** |

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

## E  Code examples

We now show how our dispatch rules are implemented in CoLA. Specifically we show the implementation of our inverse rule.

```python
from plum import dispatch

@dispatch
def inverse(A: LinearOperator, **kwargs):
    kws = dict(tol=1e-6, P=None, x0=None, pbar=False, info=False,
        max_iters=5000)
    kws.update(kwargs)
    method = kws.pop('method', 'auto')
    if method == 'dense' or (method == 'auto' and np.prod(A.shape) <=
        1e6):
        return A.ops.inv(A.to_dense())
    if issubclass(type(A), SelfAdjoint[Sum]) and (method == 'svrg' or
        (method == 'auto' and len(A.A.Ms) > 1e4)):
        return SymmetricSVRGInverse(A.A, **kws)
    if issubclass(type(A), Sum) and (method == 'svrg' or (method == '
        auto' and len(A.Ms) > 1e4)):
        return GenericSVRGInverse(A, **kws)
    if issubclass(type(A), SelfAdjoint) and (method == 'cg' or (method
        == 'auto' and np.prod(A.shape) > 1e6)):
        return CGInverse(A, **kws)
    if method == 'gmres' or (method == 'auto' and np.prod(A.shape) > 1
        e6):
        return GMResInverse(A, **kws)
    else:
        raise ValueError(f"Unknown method {method} or CoLA didn't fit
            any selection criteria")

@dispatch
def inverse(A: Identity, **kwargs):
    return A

@dispatch
def inverse(A: ScalarMul, **kwargs) -> ScalarMul:
    return ScalarMul(1 / A.c, shape=A.shape, dtype=A.dtype)

@dispatch
def inverse(A: Product, **kwargs) -> Product:
    output = [inverse(M, **kwargs) for M in A.Ms].reverse()
    return Product(*output)

@dispatch
def inverse(A: BlockDiag, **kwargs) -> BlockDiag:
    return BlockDiag(*[inverse(M, **kwargs) for M in A.Ms],
        multiplicities=A.multiplicities)

@dispatch
def inverse(A: Kronecker, **kwargs) -> Kronecker:
    return Kronecker(*[inverse(M, **kwargs) for M in A.Ms])

@dispatch
```

```
887  def inverse(A: Diagonal, **kwargs) -> Diagonal:
888      return Diagonal(1. / A.diag)
889
890
891  @dispatch
892  def inverse(A: Unitary, **kwargs) -> Unitary:
893      return Unitary(A.H)
894
```