# OpenReview forum: "CoLA: Exploiting Compositional Structure for Automatic and Efficient Numerical Linear Algebra"
_NeurIPS.cc/2023/Conference — NeurIPS 2023 poster_

### Official Review · Reviewer_V5p6 · 2023-06-13

**Soundness:** 4 excellent
**Presentation:** 4 excellent
**Contribution:** 2 fair
**Rating:** 7
**Confidence:** 3

**Summary:**

 Similarly to automatic differentiation which enabled to shift the focus from computing derivatives to deriving new algorithms, the authors propose to automate large scale linear algebra, with "structure-aware" linear algebra. Various algorithms leverage problem structure to expedite the evaluation of operations on linear operators. This toolbox aims to bridge the gap, allowing individuals without extensive knowledge of algorithms and tricks to still benefit from them. The successful accomplishment of this paper's objective could have a fair impact on the machine learning community by automating a significant portion of model optimization.

**Strengths:**

This article is well written. The problem is well posed, and the proposed solution is convincing.

Importantly, the library's well-thought-out and modular structure is crucial for its growth and positive impact on the community.

**Weaknesses:**

The number of dispatch rule is relatively small. The authors mention 70 dispatch rules on `l. 224`, but in appendix A only a dozen such rules are shown (perhaps we reach 70 dispatch rules by counting all the possible combination?). If I had to write a new model, I'm wondering if (1) I would use this framework, with the (arguably small) associated overhead of using a new library or (2) simply scroll through appendix A to see what are the rules relevant for my implementation.

**Questions:**

**Porting to Julia.**

It is a nice use case of multiple dispatch, and I like the functional paradigm used. Are there any plans to port it in Julia? There already exists in the `LinearAlgebra` module "structure aware" types (`Symmetric`, `Hermitian`), and the multiple dispatch-functional paradigm are built in, so I believed it would certainly meet its public there.

**Using SVRG.**

`l. 245` and `248`, the complexity of SVRG seems bigger than Lanczos (at least in $\kappa$; what is $M$?). A reference for the convergence rate of CG and Lanczos would be welcomed. Why is an accelerated version of SVRG not considered, to speed up convergence and further improve the runtime performances? The cyanure toolbox holds some [benchmarks](http://thoth.inrialpes.fr/people/mairal/cyanure/benchmarks.html) along with some [references](http://thoth.inrialpes.fr/people/mairal/cyanure/references.html). Is it because of the memory limitations?

**Arnoldi Iterations.**

In Fig. 3, it is shown that `Arpack` performs better than the Python implementation of the Arnoldi operations. Given the modularity of your approach, wouldn't it be worth to add a dispatch rule which specifically uses `Arpack`?

**Typos.**

* `l. 82` : matrix vector **multiples** ?

* Fig. 1 could benefit from having one legend for all 3 subplots. Contrary to what is asserted in the paper's checklist, there are no error bars. Given the moderate runtime ($10^2$s), using more than 3 repetitions (e.g 10) with error bar is necessary.

* `245`: missing parenthesis

**Limitations:**

The article has no Limitations section but seems compared extensively to other libraries in sec. 3.4.

---

> ### Author Rebuttal · Authors · 2023-08-10
>
> We thank you for your thoughtful and supportive review! Below we hope to bring clarity to your questions.
>
> > If I had to write a new model, [could I] simply scroll through appendix A to see what are the rules relevant for my implementation?
>
> Yes, implementing the necessary rules and algorithms for a specific problem is always possible. However, while one could simply use Appendix A as a lookup table to select appropriate algorithms, there are many scenarios where the best rules may not be immediately obvious, especially as the compositional structure becomes increasingly nested and/or complex. Additionally, we note that—beyond the dispatch rules—our library includes memory-efficient gradients of iterative operations that are difficult to implement correctly, particularly while retaining the matrix free LinearOperator abstraction. We expect that most researchers writing their own implementations of common algorithms such as GMRES or stochastic Lanczos quadrature will not find it high on their priority list to implement memory efficient backpropagation rules and thus leave the compute and memory savings unexploited. We also would like to think that having a large existing linear algebra ecosystem (that new rules can be slotted into if necessary) will help free up researcher time for other pursuits.
>
> **SVRG:** We have listed the complexity of CG and Lanczos $O(\sqrt{\kappa}\log1/\epsilon)$ in terms of the number of matrix-vector-multiplies, equivalent to full passes through the sum when applied to a sum linear operator $A=\sum_{i=1}^M A_i$, where M here is the number of elements in the sum [7]. The $O((M+\kappa)\log1/\epsilon)$ iteration complexity of SVRG (as it is usually expressed) becomes $O((1+\kappa/M)\log1/\epsilon)$ measured by these full passes. For large values of M, SVRG can still be faster than CG/lanczos even without the acceleration.
> We found that well chosen momentum values for accelerated SVRG can result in speedups; however, compared to the learning rate, we found it difficult to automate the selection of this hyperparameter from the data in a robust and inexpensive way.
>
> **Porting to Julia:** This would definitely be a worthwhile endeavor and we hope that this can be accomplished in the future, but it remains outside the scope of our current project. We chose to write the library in Python to interface with the popular ML frameworks of PyTorch and Jax. Nevertheless, the design of our library was certainly influenced by Julia and its programming paradigm, and we hope that these ideas can make it back into Julia LinearAlgebra.
>
> **Arnoldi Iterations:** We appreciate the suggestion and wrapping ARPACK. This is something that we have been debating ourselves as there is a clear runtime benefit but at the cost of not modular, not clean and hard to read code for the user. However, for the particular Arnoldi case that you mention, the best way to proceed is simply to wrap ARPACK and the result will still be autograd enabled because of how we define our custom autograd rules.
>
> **Typos:** We will correct the typos you have identified.

---

> > ### Comment · Reviewer_V5p6 · 2023-08-11
> >
> > Thanks to the authors for their rebuttal. While not fully convinced of the added value of the dispatch rules (in place of implementing directly the efficient update rules myself), I am very sensible to the argument of providing efficient gradients for some specific algorithms which can be a very tedious process. I am convinced that an expanded linear algebra ecosystem emerging from this library could be highly beneficial for researchers. I hope this will make it to Julia LinearAlgebra and look forward to trying the library.

---

> > > ### Author Response · Authors · 2023-08-19
> > > **Thank you**
> > >
> > > We really appreciate your support and thoughtful comments, and look forward to having you as a user of CoLA! Yes, indeed, we believe efficiently backpropagating through iterative methods has the potential to jumpstart many research efforts. We will take your comments into account when updating the manuscript. If you are open to increasing your score in light of our rebuttal, it would be much appreciated, but of course no pressure.

---

### Official Review · Reviewer_ZAWv · 2023-06-29

**Soundness:** 3 good
**Presentation:** 3 good
**Contribution:** 1 poor
**Rating:** 5
**Confidence:** 4

**Summary:**

This work proposes a new library for solving linear algebra problems involving structured matrices. The library incorporates highly efficient linear algebra kernels tailored to handle specific types of structured matrices. Moreover, it uses compositional rules to tackle problems involving matrices with composition structures. By employing these rules, the library eliminates the need for manual implementation of many efficient algorithms applicable to such matrices.

**Strengths:**

1. Designing efficient numerical linear algebra libraries is important to the machine learning community.

2. This work is well-written and the presentation is clear.

**Weaknesses:**

Overall I believe the novelty of the work is very limited. The numerical linear algebra algorithms discussed in the paper do not appear to be novel, and the proposed library essentially consolidates these existing algorithms. While this library may be the first to utilize compositional rules for automated algorithmic development, the rules themselves are straightforward and uncomplicated, lacking significant novelty. In summary, this work represents a combination of various engineering efforts, but its scientific contributions are mostly incremental.

**Questions:**

n/a

---

> ### Author Rebuttal · Authors · 2023-08-10
>
> Thank you for your review. We appreciate the positive remarks regarding the clarity of our work. Below, we clarify the novel methodological contributions of our submission. At the same time, we also note that many impactful papers at NeurIPS have been centered around frameworks where algorithmic innovation is not the focus (see references in the general remarks).
>
> **Novelty in Composition Rules**: We respectfully but strongly argue that the use of compositional dispatch rules to recursively subdivide linear algebraic operations is a substantial innovation. The rules may seem “uncomplicated,” but in this context, simplicity adds to the practical utility of the approach, and we ask that our research is judged by its impact rather than its complication. (We also note that the NeurIPS reviewing guidelines explicitly endorse new combinations of known techniques.) The potential impact of this approach can be seen in Figures 1 and 2, which thoroughly demonstrate scenarios where this compositional approach is not only advantageous but also atypical.
>
>
> **Additional Contributions Overlooked**: We’d like to highlight several novel contributions of our paper that may have been overlooked. We have introduced a novel algorithm for stochastic diagonal and trace estimation of sum objects that we prove (in appendix B.1) and empirically verify that it achieves a speedup over the standard Hutchinson estimators (in the attached PDF, Figure A - Left). Moreover, we provide memory efficient autograd rules (section 3.4) to be used in conjunction with the iterative algorithms.
>
> **Significance of Software Libraries in Scientific Contribution**: We'd like to emphasize that many impactful papers presented at NeurIPS and other leading conferences have been centered on the development and presentation of software libraries. This community has recognized that the practical implementations, when done effectively, propel research forward, as evidenced by works such as [1, 2, 3, 4, 5] (see general remarks). These libraries, even without introducing radically new algorithms, provide tangible benefits to the research community.
>
> In conclusion, while individual rules may appear obvious, the framework for integrating these structures in a general way, and the cumulative impact of the many features we introduce help move the research community forward. Given these points and the precedence of NeurIPS accepting software-centric papers as crucial contributions, we kindly request you to reconsider your assessment of our work.

---

> > ### Comment · Reviewer_ZAWv · 2023-08-20
> >
> > I appreciate the comprehensive response from the authors, and I've decided to adjust my score accordingly.
> >
> > Regarding the supplementary contributions, the authors have mentioned the "Doubly stochastic diagonal and trace estimation" algorithm as a novel aspect. However, this term is absent in the paper's main body, which might leave readers confused about its relevance and the contributions of the paper.
> >
> > I agree that a good software library for scientific computations is important, even if the foundational idea seems straightforward. However, without access to the code, determining its significance based solely on the paper is challenging.

---

> > > ### Author Response · Authors · 2023-08-21
> > >
> > > Thank you for your response and reevaluating our work.
> > > We briefly mentioned our doubly stochastic estimator in the main text (lines 250-260) but indeed we would expand this discussion to highlight this contribution.

---

### Official Review · Reviewer_Wjxi · 2023-07-04

**Soundness:** 3 good
**Presentation:** 3 good
**Contribution:** 4 excellent
**Rating:** 7
**Confidence:** 3

**Summary:**

In this paper, the authors introduce CoLA, a library which streamlines the use of various linear algebra routines within frameworks of relevance for machine learning applications. The library revolves around the LinearOperator object, and provides numerous implementations of various numerical algorithms involving operations with such object, from inversion (system solution), to eigenvalue computation, to operators manipulation.

Most notable features of the library are:
- Structure-awareness and automatic dispatch: the CoLA framework is able to leverage information regarding the relevant structure of the LinearOperator considered (such as positive-definiteness, symmetry, sparsity, composition of operators), so to automatically identify and utilise the most apt algorithm specialisation for the target operation
- Integration with existing frameworks: CoLA can interface with both JAX and PyTorch codebase, supports both GPU and TPU acceleration, and has some support for low-precision arithmetic and complex numbers
- Autograd for iterative routines: CoLA implements some relevant iterative routines for linear algebra, and defines some ad-hoc rules for efficiently performing automatic differentiation through them

The main reported results pertain the performance comparison of the CoLA-implemented routines versus alternative implementations. Overall, CoLA performance is shown to be comparable to that of the baselines; moreover, when there is gain to be had from leveraging the structure of the underlying linear operator, CoLA is shown to be able to do this effectively.

**Strengths:**

- The authors propose a very useful framework. Most noticeably, it automates away the need for manually tuning the method choice depending on the properties of the operator, when numerical linear algebra algorithms are involved. Moreover, it effectively leverages some clever design choices (such as multiple dispatch). Overall, it can become a valid tool for numerous applications in the field of ML (and other fields as well)
- The authors propose an interesting solution to efficiently performing autograd on iterative procedures, which are ubiquitous in linear algebra applications
- The paper is reasonably well-written. Even though the breadth of applications considered in their work is indeed rather large, the authors still manage to present the key advantages of their framework in a clean manner, without being dispersive

**Weaknesses:**

- The main weakness I see, is the lack of actual “novelty” in the work being proposed - at least in the classical sense of the word. Apart from the autograd rule in Appendix B2, in fact, the various methods proposed and implementation choices are not new. This notwithstanding, the main goal of the project consists in collecting available linear algebra routines into a unified, ready-to-use, efficient library which can be easily encapsulated within existing ML frameworks, and as such it is still valuable to the research community

**Questions:**

Overall I’m quite satisfied with the paper. One minor doubt / curiosity I still have is:
- When comparing CoLA with other existing baselines in Fig3, it seems like your implementation underperforms in both (a) PCA and (d) Spectral Clustering. You elaborate a bit for (d) in Appendix D4, but could you expand on this? In particular, is the difference in performance simply due to the lack of an optimised implementation on your side, or are there some structural causes? And what are the main implementation differences?

**Limitations:**

Limitations of CoLA are not explicitly commented upon, but at the same time the framework proposed seems very flexible and efficient (as showcased in the experiments). The main limitations are hence connected to what methods are readily implemented in the framework, rather than being structural.

---

> ### Author Rebuttal · Authors · 2023-08-10
>
> We sincerely appreciate your thoughtful and supportive review. We were glad to see your appreciation of the broader ways in which CoLA can help contribute to the scientific community.
>
> **Performance gap on PCA and Spectral clustering**. On PCA, we believe the runtime gap is primarily the result of CoLA’s Python overhead. On the spectral clustering example we use Lanczsos to compute the smallest eigenvectors whereas scikit-learn typically uses LOBPCG. When comparing against scikit-learn using Lanczos (sk(L) vs CoLA(L)), we perform slightly better due to minor differences in the Lanczos implementation (scikit learn uses implicit restarts and we do not). We have now incorporated LOBPCG into CoLA and we compare the results (sk(B) vs CoLA(B)) in Figure B (Left), with the CoLA implementation coming out slightly ahead again.
>
> Please let us know if we can assist with any other questions.

---

> > ### Comment · Reviewer_Wjxi · 2023-08-11
> >
> > I thank the authors for addressing my doubt, which I consider resolved. I confirm the score given, and once again underline that, even though I understand the other reviewers' concern on the possible lack of novelty in this work, it is my opinion that this paper deserves being acknowledged nonetheless

---

> > > ### Author Response · Authors · 2023-08-18
> > > **Thank you**
> > >
> > > Thanks for engaging with our response. We really appreciate your support!

---

### Official Review · Reviewer_pi1f · 2023-07-07

**Soundness:** 3 good
**Presentation:** 3 good
**Contribution:** 2 fair
**Rating:** 6
**Confidence:** 4

**Summary:**

This works presents CoLA, a framework for extending the linear algebra interface of modern numerical packages to take advantage of the structural properties of linear operators present in machine learning and other applications. By adding adaptive multi-type based dispatching CoLA adaptively exploits both dense and iterative methods to decrease the computational costs of performing a given operation. CoLA is also capable of exploiting the compositional structure that is present by combining different linear operator properties, creating a large collection of possible applications. Additionally, CoLA provides all these features in an extensible framework capable of backpropagation to ensure integration in modern machine learning and deep learning applications. Evaluations support the authors assertion that providing structural information during processing facilitates better performance on many benchmark applications.

**Strengths:**

- The work clearly addresses an ongoing issue in many numerical linear algebra applications that require specialized structural operator structures to be individually implemented and exploited on a per-application basis. This process is not on tedious and time-consuming but also ripe with opportunities for error during the computation of the backward pass updates required for integration in a modern machine learning application.
- Overall the writing and exposition of the problem, proposed solution and evaluations are clear and well articulated in the text.
- The proposal dovetails naturally with the well-known LinearOperator interfaces that exist in Scipy and provide an easier route to define further extensions for developers to provide application-specific knowledge for further customization.
- An interface for composing different linear operator properties for CoLA to exploit seems to be a novel and interesting extension over other existing implementations.
- Performance results sufficiently demonstrate that having access to more structural information naturally supports more opportunities for application performance improvement while lowering the cost of the developer to exploit that structure through an intuitive and simple interface.


**Weaknesses:**

- Though interesting the work presented may be a better fit in a venue that focuses on numerical methods and software. The intended audience is quite broad.
- The core contributions and methods are relatively straightforward and are well-known to the numerical linear algebra community. This work seeks to make the process of exploiting the structural properties of linear operators easier for developers to use and integrate into a modern machine-learning application but it's not clear whether this contribution would have an appreciable impact on the developer community. Though this is my personal opinion it seems that problems encountered by developers are simple enough to be solved manually in most cases.
- The evaluation section focuses on applications that exhibit basic structure but it's hard to see the value of the additional features mentioned: backprop and lower precision.
- There seems to be a strong reliance on the appendix to fill in missing explanations due to size constraints.
- Although the interface provides more flexibility regarding basic and/or compositional structure this still leaves a host of additional options to select the appropriate iterative approach to solve a sparse matrix.

**Questions:**

- It probably doesn't help that one of the applications I know the most about, spectral clustering, is the application where CoLA seems to provide mixed results. Would closing the performance gap between CoLA and the sk (PyAMG) backend be a simple process of extending the multi-dispatch interface?
- I found the added ability to efficiently backprop through iterative solve methods interesting, were evaluation results for this feature presented in any of the experiments?


**Limitations:**

The limitations of the work are clear from the presentation and no issues require further acknowledgment, to the best of my knowledge.

---

> ### Author Rebuttal · Authors · 2023-08-10
>
> Thank you for your thoughtful review. In our response, we provide substantial clarifications, as well as experimental results catalyzed by your comments. We appreciate your feedback, and hope you can consider increasing your score in your final evaluation. We strongly believe this effort will have a significant impact on the machine learning community, comparable to highly impactful libraries such as GPyTorch [6] and BoTorch [2], both of which appeared at NeurIPS in previous years. We would be happy to engage if there are further questions.
>
> **Fit to NeurIPS**. Our library, though by no means limited to machine learning problems, was specifically designed with machine learning applications in mind. Consider the following design decisions:
>
> 1. _Machine learning specific features_. Unlike other frameworks, CoLA offers backpropagation (a nontrivial contribution of our paper), GPU acceleration, and low-precision operations—three necessary features for modern machine learning applications. While these features are broadly applicable, their impact has unquestionably been dominated by ML in recent years.
> 2. _Algorithms suited for the “implicit structure” of machine learning problems_. As outlined in Section 3.3, many of the algorithms used by CoLA (e.g. randomized diagonal estimation, randomized preconditioning, SVRG, etc.) are especially well-motivated for ML objective functions, which often feature large summations over data that are amenable to randomized algorithms.
> 3. _Ability to rapidly prototype with different structures_. The flexibility that results from the use of dispatch rules is of particular value to ML researchers who are more inclined to prototype different structures (diagonal, convolution, low-rank) without a strong a priori sense of what might provide a good approximation.
> 4. _Evaluation on machine learning applications_. The applications in the paper are dominated by machine learning relevant topics (GPs, equivariant neural nets, neural PDEs, spectral clustering, PCA…).
>
> We thus argue that the machine learning community has the most to gain by using CoLA. We would also note that, as stated above NeurIPS has been a venue for similar software frameworks like GPyTorch [6] and BoTorch [2]. CoLA is similar in nature to these other frameworks, but has arguably an even greater potential for impact in ML as its applicability spans more applications.
>
> **Impact on the developer community**. We respectfully disagree that “problems encountered by developers are simple enough to be solved manually in most cases.” In machine learning applications (e.g. second order optimization), it is often common to prototype with different types of matrix structure (e.g. block diagonal versus low-rank versus Kronecker approximations of the Hessian matrix). While the rules that govern these different structures are not necessarily complicated, switching between these different structures is a tedious and error-prone process. Indeed, this is a pain point that we encountered in many of our own projects, which inspired our development of this library. CoLA will automate away this process, enabling more rapid prototyping.
>
> The usefulness of such automation should not be underestimated. Automatic differentiation frameworks have significantly impacted ML research and development, even though computing gradients is conceptually straightforward. We argue that CoLA will have a similar effect by targeting a different bottleneck in the ML pipeline.
>
> **Value of backprop and lower precision.** We note that these two features are critically important to our library. Optimization through backpropagation is now the dominant paradigm in machine learning, and low-precision arithmetic is becoming increasingly prominent to improve speed and memory consumptions. We also emphasize that our implementation of these features are nontrivial contributions. A naive implementation of backpropagation (i.e. directly backpropagating through numerical methods) would incur significant memory (see Figure 4 in the supplementary), and the standard implementations of numerical methods are well known to be unstable for low-precision arithmetic.
>
> For a specific experiment that demonstrates the value of backpropagation, we would draw your attention to the Gaussian process application in Figure 3. The parameters of the kernel are chosen by backpropagating the negative log marginal likelihood function, which requires computing a solve and log determinant of the kernel matrix, as well as the gradients of these operations. To demonstrate the value of low precision, we added a linear regression experiment in float16 to show the runtime efficiencies that can be gained. Please see Figure B (Right) of the attached rebuttal pdf.
>
> **Selecting the appropriate interactive approach**. We believe that there may be some confusion about how the appropriate numerical method is selected. Most users will likely use the *high-level CoLA interface* (i.e. calling `cola.solve`, `cola.eigs`, `cola.trace`, etc.), in which CoLA automatically determines an appropriate default numerical method based on the underlying structure and size of the linear operator. “Power users”, who may want to specify the underlying algorithm, can use the *low-level CoLA interface* (i.e. directly calling `cola.cg`, `cola.gmres`, etc.) or pass a keyword argument to the high level interface solve(A, b, method=”cg”). We will clarify this point in the paper.
>
> **Performance of CoLA on Spectral clustering**.  As you note, there is a performance gap between CoLA and scikit-learn on the spectral clustering application. This gap is the result of CoLA not using the same algorithm as scikit-learn. We have now incorporated LOBPCG into CoLA and you can find the results on the attached PDF, Figure B (Left). Now that we have included LOBPCG (sk(B) vs CoLA(B)), CoLA again achieves better runtimes.
>
> We appreciate your thoughtful questions and we are happy to engage further!

---

> > ### Comment · Reviewer_pi1f · 2023-08-16
> >
> > I thank the authors for their thorough responses. Based on their feedback I have increased my rating for the paper accordingly. The methods proposed will benefit the wider machine learning community.

---

> > > ### Author Response · Authors · 2023-08-18
> > > **Thank you**
> > >
> > > Thanks for engaging with our response and increasing your score. We really appreciate your support!

---

### Official Review · Reviewer_qUas · 2023-07-07

**Soundness:** 3 good
**Presentation:** 3 good
**Contribution:** 2 fair
**Rating:** 5
**Confidence:** 3

**Summary:**

The paper presents a library to automate the efficient execution of numerical linear algebra kernels commonly occurring in ML applications. The library recursively exploits compositional structure beyond versus standard packages which treat numerical matrix kernels as back boxes. The proposed framework provides memory efficient automatic, differentiation, low precision computation, GPU acceleration in both JAX and PyTorch.

**Strengths:**

-) The quality of the manuscript is good. All concepts are communicated clearly and the paper is very well-written.
-) The authors have put a lot of effort to provide automation for several important numerical kernels and structures. A long list of important applications are listed, and COLA can be of major significancy in several ML areas, essentially speeding up innovation.
-) The numerical results indicate that COLA can be faster than baseline alternatives on a wide range of numerical tasks.

**Weaknesses:**

-) One aspect I found confusing is the lack of information regarding the numerical algorithms featured in some of the results. For example, in Figure 1, it is not clear why the library is faster for the Kronecker problem. Likewise, the same is true for the Bi-Poisson problem. Can the underlying algorithms be found online?
-) In similar spirit, it is not clear whether COLA is compared against state-of-the-art numerical approaches. For example, Multigrid is the de-factor choice for elliptic Poisson problems, is this what the authors list as 'iterative'? If not, what is the point of listing a comparison against a non-optimal iterative (or direct) algorithm?
-) In Figure 3, scipy is competitive with COLA, if not better in some tasks. This furthers complicates the message of the paper. What is the main reason for publication? The fact that COLA includes a wide-range of solvers for composite tasks or that it can be faster or more memory efficient in general?

**Questions:**

-) Is there any new numerical method involved in the library? My understanding is that every single numerical method used for linear systems, eigenvalue problems, etc, is known, and the main novelty of the paper is to wrap them together in an efficient manner for composite tasks.
-) Judging COLA as a library requires some reasonable level of insight in the library itself. I understand that the authors must stay anonymous, but it is rather difficult to fully understand the benefits of COLA without looking at the code itself.

---

> ### Author Rebuttal · Authors · 2023-08-10
>
> We are appreciative of your thoughtful feedback. In our response, we provide important clarifications, and new results inspired by your comments. Although contributions of this type can be hard to evaluate, we believe CoLA should be judged by its strong potential for scientific impact, and hope you can consider raising your score in light of our response.
>
> **Why CoLA composition rules produce algorithmic speedups and Figure 1**. The main objective of Figure 1 is to show how exploiting the compositional structure provides an algorithmic improvement. For example, for matrices with Kronecker product structure, it is common practice to use an iterative algorithm like CG to perform a linear solve (see e.g. [6]). However, we show it is more efficient to split the problem into two using the Kronecker structure: in particular, decompose $(K_T \otimes K_X)^{-1} \mathrm{vec}(Y) = \mathrm{vec}(K_T^{-1} Y K_X^{-1})$ and the complexity is reduced from $O\big(\sqrt{\kappa_T \kappa_X}(m^2n + mn^2)\log1/\epsilon\big)$ to $O\big(\sqrt{\kappa_T}m^2n\log1/\epsilon+\sqrt{\kappa_X}mn^2\log1/\epsilon\big)$. The computational burden is likewise reduced for when using dense LU or Cholesky based solvers.
>
> For the BiPoisson problem, ($\Delta^2 x = \rho$), it is more efficient to use the product structure and perform the two linear solves separately. As you mention, a specialized multigrid method can very efficiently solve the discretized elliptic differential operator here. Just as in the dense and iterative approaches that we discussed, the multigrid method also benefits from splitting up the problem with CoLA’s composition rules. We have run the multigrid method both with the CoLA decomposition (solving the PDE by inverting $\Delta$ twice) and without (solving by inverting $\Delta^2$) and show in the attached PDF (Figure A (Right)) that applying that doing so yields significant runtime improvements. In other words, CoLA’s approach of recursively breaking up structure provides runtime benefits independent of the algorithm being used to solve the problem (CG or multi-grid).
>
> **Why CoLA should be published at NeurIPS**. The research behind CoLA serves to substantially reduce the bottleneck of deriving efficient algorithms that exploit the structure commonly found in machine learning and scientific computing. Regarding the scientific impact of frameworks such as CoLA, we would like to draw an analogy to PyTorch and reverse mode automatic differentiation (NeurIPS 2019). From a narrow point of view, autograd is merely an application of the chain rule, and yet its impact on machine learning research has been almost immeasurable. Autograd obviates the need for deriving backpropagation rules for each model separately, and bespoke autograd rules (such as our own for iterative solvers in CoLA) can be slotted into an existing language of differentiable functions only when necessary, without requiring the whole structure to be constructed anew. Likewise with CoLA, we have developed an approach such that new rules can be slotted into an existing linear algebra ecosystem, and that ecosystem need not be rewritten for each use case.
>
> The potential impact of this framework is extremely large. In a sense, machine learning is largely linear algebra, and common modeling assumptions give rise to structure that can be exploited for significant computational savings. CoLA will help make researchers more broadly aware of the structure they can exploit, and significantly reduce the bottleneck to implementing methods that exploit structure, as well as prototyping various structures for their problems. The speed gains when using CoLA depend on the degree of compositional structure. For some problems like the Schrodinger equation (Figure 3 (e)), CoLA achieves parity with SciPy; in more complex problems like equivariant neural networks (Figure 1(c)) CoLA achieves remarkable speedups over existing solutions by exploiting compositional structure (see Figure 1). Of course, CoLA will not be a magic bullet for every problem, and we believe it is actually to the paper’s credit that it provides an honest and comprehensive presentation, including results where CoLA is essentially on par with alternatives.
>
> **On the purpose of Figure 3**. We would like to clarify the purpose of Figure 3. While Figures 1 and 2 demonstrate the efficiency gains on problems with compositional structure, Figure 3 demonstrates the breadth of CoLA’s applicability in real-world applications, including on problems with no compositional structure. On these problems without compositional structure, we do not expect CoLA to outperform existing specialized methods. However, as seen in Figure 3, CoLA remains competitive with these specialized methods, demonstrating that even in the “worst case” scenario (no compositional structure) there is no downside to using CoLA. We will clarify this point about Figure 3 in the main text.
>
> **Regarding new numerical methods**, we believe you may have missed the novel doubly stochastic trace / diagonal estimator that we introduce Section 3.3. This algorithm, though not the centerpiece of our paper, is an important methodological contribution as it yields lower variance than the standard Hutchinson estimator (see Appendix B.1). Moreover, we provide a novel procedure to automatically compute gradients through iterative methods, as well as an automatic procedure to compute diagonals and transposes / adjoints of linear operators (Sections 3.1 & 3.4) through their matrix-vector product routine.
>
> **CoLA’s code**. We have been careful to retain anonymity, but we are beyond excited to publicly announce the library, as we feel the community would value it greatly. However, we argue that the key ingredients behind CoLA (a pleasingly simple programmatic mechanism for exploiting compositional structure, as well as the algorithms covered in Section 3) are sufficiently general purpose concepts that can be evaluated independent of implementation.

---

> > ### Comment · Reviewer_qUas · 2023-08-19
> >
> > The rebuttal (.pdf) is useful and so are the responses.
> >
> > Some responses:
> >
> > -) There is nothing surprising from a numerical perspective and I do not think that what the authors think as novel really is. The break-up of most if not all structures discussed in the paper are basically trivial for anyone working in NLA. Numerical analysis is not the main part of the paper so I am not going to reject just for that, but I would pale down the tone.
> >
> > -) The overall framework is useful and I am in favor of it. Nonetheless, judging the full potential of a library-based framework without running /reviewing the code and having access to more information is (extremely) limiting regardless of what papers where accepted in the past. Stating that the whole framework is a simple programmatic mechanism is also not relevant. This is a general issue with software-oriented papers and double blind peer review.
> >
> > -) The doubly stochastic trace estimator (I did not miss it) is rather straightforward. My understanding is that you eliminate the need to consider the cross-product of the matrix sum from the variance upper bound. But how often do you really encounter trace computations where 'A' is expressed as the sum of 'm' matrices? Most of the times we want to compute the trace of f(A) where f(x)=x^3, f(x) = e^x, etc (triangle counting, subgraph centrality). How do you break this into pieces to fit your framework? What non-trivial application exists for the proposed diagonal estimator?
> >
> > -) The statement "CoLA will not be a magic bullet for every problem, and we believe it is actually to the paper’s credit that it provides an honest and comprehensive presentation" is rather strange. Is there any other way to write a paper other than provide an honest assessment? I think you meant to say that you did the best of your ability to present a fair and informative comparison even in scenarios where CoLA is not expected to outperform.
> >
> > Overall, the rebuttal is useful and I will increase the score to borderline accept. Good luck with your submission.

---

> > > ### Author Response · Authors · 2023-08-21
> > > **Clarifications**
> > >
> > > Thank you for your response! We appreciate your support. Below we make some clarifications in response to your comments.
> > >
> > > * We agree that breaking-up structure is a well-known concept in NLA. Yet, the novelty in our paper does not come from being the first ones to exploit the structure but by doing it automatically (through our recursive dispatch rules). All the rules in appendix A are simple but they still require that the practitioner write an explicit method to use the given structure at hand, and instead we provide a framework to do so automatically. Let us illustrate with an example. If a user wants to compute the determinant of a matrix $A=B \otimes C$, where $A$ is a tridiagonal matrix, and $C=PLU$ is the product of P,L,U matrices in its PLU decomposition. Our framework will split the determinant into $\mathrm{det}(A)^{dim(C)}\mathrm{det}(B)^{dim(B)}$, compute the efficient diagonalization of a tridiagonal matrix to find $\mathrm{det}(A)$, extract the diagonal of $L$ and $U$, and compute the sign of the permutation $P$ to find its determinant and then assemble all these components together into the final result. This functionality is highly practical, as it helps free the user to focus on the modeling assumptions behind A rather than on the linear algebra. There is no current framework with these capabilities.
> > >
> > > * Our doubly stochastic estimator can be highly practical. While the classic application of stochastic trace estimation is matrix polynomials, more recent work from the ML community applies this technique to matrices that are summations over datasets [1,2]. Consider computing the diagonal or trace of the Hessian of a neural network. This diagonal is relevant in quantization (where in e.g. [1] it is computed using the naive Hutchinson estimator), in optimization [2], and elsewhere. Since the Hessian is the sum over hessians for a large number of data points ($m$), we can expect a reduction in the number of iterations required to reach the same variance by roughly $1/m$.
> > >
> > > * Indeed we phrased that poorly. As you mentioned, our goal is to present a fair and informative comparison even in scenarios where CoLA is not expected to outperform.
> > >
> > > Thank you again for your feedback.
> > >
> > > [1] Dong, Zhen, et al 2020. "Hawq-v2: Hessian aware trace-weighted quantization of neural networks." NeurIPS.
> > > [2] Liu, H., et al. 2023. Sophia: A Scalable Stochastic Second-order Optimizer for Language Model Pre-training. arXiv 2305.14342v1.

---

### Author Rebuttal · Authors · 2023-08-10

We thank the reviewers for their thoughtful and strongly supportive feedback. In this general post, we highlight some of the new experiments that we conducted inspired by reviewer comments, and provide some general remarks about CoLA. We also have separate posts individually replying to each reviewer.

We were happy to see that reviewers share our enthusiasm for CoLA. Linear algebra is a core foundation for machine learning algorithms, where common modeling assumptions give rise to structure that can be exploited for significant computational savings. CoLA will help make researchers more broadly aware of the structure they can exploit, and significantly reduce the bottleneck to implementing methods that exploit structure, as well as easily prototyping various different structures for their problems (as it is often not clear a priori what structure will be most beneficial for a given problem).

**Impact and significance.** An analogy with PyTorch and reverse mode automatic differentiation is helpful for understanding the significance of CoLA and potential impact. From a narrow point of view, autograd is merely an application of the chain rule, and yet its impact on machine learning research has been almost immeasurable. Autograd obviates the need for deriving backpropagation rules for each model separately, and bespoke autograd rules (such as our own for iterative solvers in CoLA) can be slotted into an existing language of differentiable functions only when necessary, without requiring the whole structure to be constructed anew. Likewise with CoLA, we have developed an approach such that new rules can be slotted into an existing linear algebra ecosystem, and that ecosystem need not be rewritten for each use case. In this respect, the simplicity of CoLA is a strength, and will help enhance its usefulness to the community. Overall, the NeurIPS and broader ML community has found software libraries that simplify the research process as significant and impactful contributions (for example, consider the NeurIPS papers [1, 2, 3, 4, 5]).

**Methodological contributions.** We also note that, while they are not the primary focus, we propose novel numerical algorithms that are important to our framework, such as doubly stochastic trace and diagonal estimation. When applied to linear operators with sum structure, we prove that this estimator has considerably lower variance than the standard Hutchinson estimator in Appendix B.1. More broadly, we have provided a novel procedure to automatically compute gradients through iterative methods, as well as an automatic procedure to compute diagonals and transposes / adjoints of linear operators (section 3.1 & 3.4) through their matrix-vector product routine.

**Additional experiments.** Inspired by reviewer feedback, we have put a significant effort into providing some new results (see attached pdf for Figures A, B and C):

 - Figure A (left) shows how our doubly stochastic estimator reduces the runtime by orders of magnitude when applied to large sums such as when estimating the variance (diagonal of the covariance) of the PCA application from Figure 2a.
 - Reviewer qUas suggested, for the Bi-Poisson problem in Figure 1(b), that we perform the comparison with a multi-grid solver, a method which is generally faster for solving elliptic PDEs. We have added this comparison in Figure A (right) using the CoLA decomposition rules to split the solve into two multi-grid solves, which like for our previous case in Figure 1 (b) when we were using conjugate gradients, CoLA also accelerates the convergence of multi-grid.
 - Moreover, for the spectral clustering example in Figure 3 we have now incorporated the LOBPCG algorithm into CoLA (before LOBPCG was giving scikit-learn an edge over us). As you can see in Figure B (left), CoLA’s LOBPCG results have improved significantly.
 - Furthermore, on Figure B (right), we have added an example on how low precision can improve runtime for linear regression.
 - Finally, we added on Figure C the runtime and memory consumption of backpropagating through a log determinant. We compare CoLA’s autograd rules against naively backpropagating through the iterative algorithm used to estimate the log determinant. The plot shows the substantial savings in runtime and memory from our approach. Taken together, Figure C and Figure 4 give the quantitative impact of our backprop rules for the two operations (solves and log determinants) needed for training Gaussian processes. As such, these backprop rules were used for the experiments in Figure 1 (a) and Figure 3 (c).

We are thankful for the questions, and would appreciate it if our responses and clarifications can be considered in your final assessment.

_References_

[1] Paszke et al., 2019. PyTorch: An Imperative Style, High-Performance Deep Learning Library. NeurIPS.

[2] Balandat et al., 2020. BoTorch: A Framework for Efficient Monte-Carlo Bayesian Optimization. NeurIPS.

[3] Daxberger et al., 2021. Laplace Redux – Effortless Bayesian Deep Learning. NeurIPS.

[4] Frank et al., 2021. Cockpit: A Practical Debugging Tool for the Training of Deep Neural Networks. NeurIPS.

[5] Pineda et al. 2022. Theseus: A Library for Differentiable Nonlinear Optimization. NeurIPS.

[6] Gardner et al., 2018. GPyTorch: Blackbox Matrix-Matrix Gaussian Process Inference with GPU Acceleration. NeurIPS.

[7] Golub et al., 2018. Matrix Computations. 4th Edition. The Johns Hopkins University Press.

---

### Decision · Program_Chairs · 2023-09-21

**Decision:**

Accept (poster)

**Comment:**

This is a well-written paper presenting work on a linear algebra library that automatically exploits various types of structure present in linear operations. It ought to be of interest to users and numerical library implementors alike. Expert reviewers all found the paper well-written and recommend acceptance to varying degrees. I second this recommendation overall.

The feedback during review can be decomposed roughly into (a) feedback on the framework itself (CoLA), and (b) feedback on the scientific contribution of the paper that presents CoLA.

For the framework itself, reviewers largely agree that CoLA seems interesting and useful. Some noted strengths include portability atop Pytorch and JAX, a variety of numerical procedures and operator structures covered, and automatic differentiation.

Regarding the scientific contribution of the paper, I would highlight the discussion between the authors and reviewers qUas, pi1f, and V5p6. These reviewers all contended that on the algorithmic side, the framework does not extend what is already known in numerical linear algebra and could be implemented on a per-application basis. As the discussion went, there is more substantial contribution elsewhere, e.g. on automation (both in kernels and in automatic differentiation) and evaluation on machine learning applications.